



# Blending 2D topography images from SWOT into the altimeter constellation with the Level-3 multi-mission DUACS system

Gerald Dibarboure[1], Cécile Anadon[2], Frédéric Briol[2], Emeline Cadier[2], Robin Chevrier[2], Antoine Delepoulle[2], Yannice Faugere[1], Alice Laloue[2], Rosemary Morrow[3], Nicolas Picot[1], Pierre Prandi[2], Marie-Isabelle Pujol[2], Matthias Raynal[1], Anaelle Treboutte[2], Clément Ubelmann [4]

[1]Centre National d'Etudes Spatiale (CNES), Toulouse, France
[2]Collecte Localisation Satellites (CLS), Ramonville St Agne, France
[3]Laboratoire d'Etudes en Géophysique et Océanographie Spatiales (LEGOS), Toulouse, France
[4]Datlas, Grenoble, France

*Correspondence to*: Gerald Dibarboure (gerald.dibarboure@cnes.fr)

**Abstract.** The Surface Water and Ocean Topography (SWOT) mission delivers unprecedented swath altimetry products. Despite SWOT's 2D coverage and precision, its Level-2 products suffer from the same limitations as their counterparts from nadir altimetry missions. Level-2 products are designed in a standalone ground-segment to meet the mission's primary science objectives. In contrast, some research domains and applications require consistent multi-mission observations such as the Level-3 products provided by the Data Unification and Altimeter Combination System (DUACS) for almost 3 decades, and with 20 different satellites. In this paper, we describe how we extended the Level-3 algorithms to handle SWOT's unique swath-altimeter data. We also illustrate and discuss the benefits, relevance, and limitations of Level-3 swath-altimeter products for various research domains.

## 1    Introduction

The Surface Water and Ocean Topography (SWOT) satellite is an international collaboration between the National Aeronautics and Space Administration (NASA, United States of America), the Centre National d'Etudes Spatiales (CNES, France), The Canadian Space Agency (CSA) and the United Kingdom Space Agency (UKSA). The mission's objective is to make the first global survey of Earth's surface water: ocean surface topography, inland water heights, river discharge… Morrow et al. (2019) or Fu and Rodriguez (2004) detail the mission's goals, and Fu et al. (2024) illustrate the breakthrough provided by SWOT during its first months of operations.



To achieve this goal, the main instrument of the SWOT mission is a Ka-band interferometer (KaRIn) that delivers a bi-dimensional (2D) view of the water surface topography (Peral et al., 2024). To meet its requirement, the interferometer requires an extremely precise attitude and orbit control system (AOCS), as well as the full suite of "conventional" altimetry payload: a nadir altimeter (POSEIDON-3C, Jason-class), precise orbit determination sensors (DORIS positioning system, precise Global Navigation Satellite System, laser reflector array) and a microwave radiometer (to correct for the wet troposphere path delay). SWOT was first operated on a 1-day repeat orbit for a few months during its calibration and validation phase. During this phase, it provided not only the first 2D water surface topography over all water surfaces (ocean, rivers and lakes, sea and land-ice), but also the first daily revisit of surface water topography by any satellite altimeter mission. Then, in July 2023, the satellite was moved on its so-called Science Orbit where it provides global coverage up to a latitude of 78° with an exact repeat of 21-days (Lamy et al, 2014).

SWOT has various product levels that follow the classical conventions from radar altimetry. Firstly, Level-0 is the reconstructed, unprocessed instrument and payload telemetry data at full resolution. Level-1B is the Level-0 data, processed to sensor units, and corrected for instrumental and geometrical effects (e.g. calibrated altimeter waveform or KaRIn interferogram). Level 2 is the Level-1 data transformed into geophysical units, and corrected for environmental effects. Level-3 is a multi-sensor calibrated (e.g. using a reference altimeter) and simplified product. Lastly, Level-4 is a multi-mission product where all datasets are mapped on uniform space-time grid scales.

The two important SWOT features over the ocean are its ability to provide a synoptic 2D view of the ocean surface without interpolation, and its precision (millimeter-scale noise). When combined, these assets make it possible to capture many ocean features such as mesoscale or internal waves, as small as a few kilometers. Figure 1 illustrates this breakthrough with a comparison between SWOT and operational multi-altimeter products. Panel (a) shows the 0.25° Level-4 altimetry from the nadir altimetry constellation (7 satellites). Ballarotta et al (2019) have shown that the effective resolution of these products is 200 km or more (in wavelength), and that a significant amount of temporal smoothing (20 days or more) is necessary to assemble such a gridded product from multiple 1D altimetry profiles. In comparison, SWOT's image provides a synoptic and precise view of the mesoscale field for this Gulf Stream region with each revisit: large eddies that were resolved in panel (a) are consistently captured by SWOT, but they are much better outlined with stronger anisotropic amplitudes and gradients, and are no longer artificially smoothed in space or in time. SWOT also resolves eddies as small as 10 km in diameter that were never observed in 2D before this mission.

At the time of this writing, the mission's error budget has not been formally published (this will be done at the end of SWOT's validation phase by the end of spring 2024). Nevertheless, preliminary reports such as Raynal et al. (2023), Bohé et al. (2023), Chen et al. (2023), Fjörtoft et al. (2023) highlight a nominal behavior of the first Level-2 products of the SWOT mission. Similarly, Peral et al. (2024) provide an in-depth review of the good behavior of the KaRIn instrument. So in the context of



this paper, we will assume that SWOT is meeting its requirements and that Level-2 products are trustworthy and final, although some minor changes might still be implemented by the SWOT Project in the coming months.

Nevertheless, the Level-2 (L2) products from SWOT suffer from the same limitations as other altimetry missions. Firstly, the processing standards (e.g. geophysical corrections and models) were defined a couple of years before launch, and there is often a significant delay for the complex ground segment to integrate state-of-the-art upgrades from research papers. Secondly, the

L2 ocean products use a complex series of quality flags that do not always match user needs, let alone be consistent with the flagging strategy of other satellites. Thirdly, the SWOT ground segment uses the Level-2 calibration strategy described by Dibarboure et al. (2022). This implementation of a SWOT standalone ground segment is consistent with the mission's hydrology requirements during the Science Phase, but it was shown to be suboptimal for oceanography, as well as a prone to multi-mission biases that may be challenging to overcome for some users.

To illustrate, the assimilation of altimetry data into ocean models for operational oceanography (e.g. Le Traon et al., 2019) require a continuous set of high-quality and consistent observed products (e.g. Davidson et al, 2019), that must be tackled within a multi-satellite, multi-agency system (e.g. CEOS, 2009; International Altimetry Team, 2021). Various models of the Ocean Predict community assimilate Level-3 altimeter products from the multi-mission DUACS system, presented by Dibarboure et al. (2011) and updated by Taburet et al(2019), Faugère et al. (2022), or Pujol et al. (2023). The system has been

operated for CNES and/or the Copernicus Marine Service and the Copernicus Climate Service for almost 3 decades and 20 satellites. However, all of them were traditional mono-dimensional (1D) altimeters.

In that context, the objective of this paper is to give an overview of the 2D Level-3 algorithms and products developed specifically for SWOT/KaRIn, as an extension of the operational DUACS system. Section 2 gives an overview of the input data as well as an overview of the L3 algorithm sequence, and section 3 gives more details about each algorithm layer. The

Level-3 product and its validation are presented in section 4. Lastly, section 5 discusses and illustrates the relevance and limitation of this dataset for various research domains.

## 2    Input data and processing overview

### 2.1    Input data

In this study, we use two types of SWOT Level-2 products. The first one is the so-called Geophysical Data Record of the nadir

altimeter instrument (i.e. traditional 1D altimeter), and the second one is the Low Resolution (LR) Sea Surface Height (SSH) product of the KaRIn swath instrument. Section 10 details the data availability and the associated documentation. For both nadir and KaRIn sensors, we use the recommended corrections and reference models from the SWOT Project documentation. For KaRIn, we use the Sea Surface Height Anomaly (SSHA) variant 2 (model-based corrections), as some corrections of the



variant 1 are not mature yet. In addition to the product documentation, Raynal et al. (2023), Bohé et al. (2023), and Chen et al.
(2023) provide a clear practical guidance on the product content and caveats.

During the calibration process of section 3.3, we combine SWOT data with the along-track Level-3 altimeter products from
the Copernicus Marine Service / DUACS (see Pujol et al. (2016) or Taburet et al. (2019) for more details). More specifically,
we use the following satellites: Sentinel-6 / 3A / 3B, SARAL, CRYOSAT-2, HaiYang-2B.

Furthermore, to get a background view of the conventional altimetry 2D ocean topography in some figures, we also use the
gridded Level-4 maps from the same DUACS system: Ballarotta et al(2019) have reported that they provide an accurate view
of medium to large ocean mesoscales. Lastly, in some sections, we make some comparisons between KaRIn and maps of
chlorophyll concentration or sea surface temperature from the Copernicus Marine Service (see details in section 10).

## 2.2   Overview of the L3 Processing sequence

In this study, we used an extended 2D version of the Level-3 algorithm sequence presented by Dibarboure et al (2011) and
updated by Taburet et al. (2019) or Pujol et al. (2023) among others: more details are also provided by the Copernicus Marine
Service operational documentation (see section 10). Figure 2 gives an overview of the SWOT algorithm sequence, which has
two separate components.

The first part is the left-hand side grey block, where we ingest the 1D nadir altimeter product from SWOT plus all other nadir
altimeters into the classical DUACS/L3 sequence. In this sequence, SWOT Level-2 products are harmonized with other
satellites (e.g. geophysical corrections, models, mean sea surface…) in order to get consistent and state-of-the-art datasets for
all sensors. More specifically, we use the so-called DT-2024 standards defined by the Copernicus Marine Service,
EUMETSAT and CNES in the ongoing reprocessing of 30+ years of altimetry (see Kocha et al (2023) and appendix from
section 9). Then the rest of the nadir Level-3 algorithms (Dibarboure et al, 2011; Taburet et al., 2019) are used in sequence:
detection of spurious measurements (also known as editing step), calibration on the reference altimeter (here, Sentinel-6),
reduction of the Long Wavelength Errors (or LWE) to mitigate residual biases between all sensors.

At the end of the nadir sequence, the 1D SWOT nadir altimeter is integrated and consistent with other altimetry satellites. All
the Level-3 nadir products are then used as an input for the second part of the SWOT processing (right-hand side grey block),
where we integrate KaRIn 2D images into this multi-mission framework.

As for the nadir altimeter, we align KaRIn's geophysical corrections with the DT-2024 standards (detailed in section 3.1), then
we edit out spurious or very suspicious pixels (specific 2D editing detailed in section 3.2). The data-driven calibration is then
activated to mitigate KaRIn systematic errors and to make the 2D images more consistent with other sensors (detailed in section
3.3). Then KaRIn images go through a noise-mitigation algorithm (detailed in section 3.4) before SSHA derivatives such as
geostrophic velocities and vorticity can be computed (detailed in section 3.5). Finally, the Level-3 products from the KaRIn



and nadir altimeters of SWOT are blended into a single product where the nadir altimeter is the center column between the
two ribbon-shaped 2D images of KaRIn.

Figure 3 illustrates the results at various steps of the processing sequence for an arbitrary zone in the Gulf Stream. In panel (a),
the input uncalibrated SSHA from the Level-2 product is used. In this example, SWOT's systematic errors can be as large as
tens of centimeters, i.e. hiding the SSHA signal of interest. In panel (b), the Level-3 calibration has been applied and the
systematic errors (e.g. image tilt from spacecraft roll) are mitigated: the actual SSHA emerges with clear mesoscale eddy
signatures and a good consistency between the ascending and descending images. However, various artifacts remain in the
image (e.g. blue spots near 32.5°N). These specific artifacts are caused by heavy rain cells that impact the Ka-band signal.
Panel (c) shows the SSHA once the Level-3 editing process is applied to take out spurious or suspicious pixels. Panel (d) is
when we applied the noise-mitigation algorithm: on this panel, the difference might seem small but section 3.4 will illustrate
that even KaRIn's millimeter scale noise is amplified in the spatial derivatives. The noise-reduction algorithm makes it possible
to retrieve very small features in panels (e) and (f).

## 2.3   Output data

The Level-3 products generated by this sequence have three variants, aligned with the Level-2 nomenclature. The 'basic'
variant is the most lightweight and simple to use, as it contains only a pre-made SSHA at 2-km resolution, that is directly
usable by oceanographers. In contrast, the 'expert' variant contains each individual Level-3 layer (each processing step from
the KaRIn block in Figure 2). The general objective for the expert product is to have a sandbox product so that the SWOT and
altimetry communities can investigate the different processing steps in more detail. More specifically, the rationale is threefold:

- Firstly, the Level-3 algorithm is a research-grade product that aggregates state-of-the-art algorithms and corrections from
  various groups. These components may need to be evaluated separately by an expert community. This can be done with
  the 'expert' variant as the Level-2 and Level-3 products can be combined to evaluate only some components of the L3.
To illustrate, some processing steps such as the calibration might be deemed unnecessary or even detrimental to some
  users, and the 'expert' variants makes it possible to customize the L3 content to each domain.

- Secondly, various research groups may be able to develop better alternatives to some Level-3 layers and the expert product
  gives a simple medium for them to test their algorithm and to evaluate the strengths and weakness independently from
  other Level-3 layers. In other words, the expert L3 is a simple testbed to evaluate future algorithms and corrections.

- Thirdly, the expert product can contain duplicate/alternative layers with experimental algorithms that are operated globally
  for large-scale evaluations. To illustrate, the 'basic' variant must contain a single default state-of-the-art correction for
  tides, whereas the 'expert' variant can integrate one or two extra tides models, that are under evaluation by the community.
  With this pre-made Level-3 with multiple models, any Level-3 user will be able to evaluate the strengths and weakness of





the experimental tide models even if they are not a tide expert, let alone be able to operate or run the tide models. The L3
user-feedback then makes it possible to consolidate how the L3 standards should evolve in the future.

To summarize, the 'basic' L3 variant is aimed at studies for researchers who do not want to get into the details of SWOT algorithms or mission-specific biases: they can get an off-the-shelf SSHA for thematic studies. In contrast, the 'expert' variant is aimed at altimetry algorithm and correction experts who do want to get into the technicalities of SWOT and altimetry standards.

Figure 2 also shows a third variant: the 'unsmoothed' Level-3 product, which is derived from the eponymous Level-2 product of the ground segment. This variant leverages the 250-m resolution of the KaRIn Low-Resolution product, whereas the basic and expert are built from the 2-km products. Strictly speaking, this is a third processing block since the input L2 for KaRIn is different. However, in practice, the algorithm sequence is the same: the only difference is that the input 'unsmoothed L2' must be completed with many missing parameters or geophysical corrections that are not provided by the ground segment at 250-
m. The rationale for the 'unsmoothed' Level-3 product is to provide a simple 'off-the-shelf' 250-m product with all the necessary content for coastal, geodesy, or sea-ice users. To achieve this, we keep only critical content from the 250-m product and we add certain geophysical models that might be needed for future standard upgrades, and then we add our Level-3 layers. Finally, we use a more convenient product structure that is similar to the 'basic' and 'expert' formats, whereas the Level-2 'unsmoothed' format was designed by the SWOT project for technical and instrument-oriented studies.

**2.4    Level-3 product versions**

At the time of this writing, the Level-3 product is available in two versions: v0.3 and v1.0. Version 0.3 was made public on AVISO in December 2023 and the more up-to-date Level-3 version 1.0 was released is May 2024 (see section 10). For the sake of clarity in this paper, and because many 'beta' glitches are now fixed, we describe only the v1.0 Level-3 algorithms, although this paper contains some original pictures from 2023 and v0.3. An overview of the differences between v0.3 and v1.0
is given in the appendix of section 9.

**3    Level-3 step-by-step algorithm description**

**3.1    Updating L2 algorithms**

The Level-2 products from SWOT were defined approximately two years before launch. They integrated state-of-the-art corrections (e.g. sea-state-bias, tides models) and reference surfaces (e.g. mean sea surface, mean dynamic topography) at the
time. But the state-of-the-art has evolved since then. More generally, each satellite has some periodic standards upgrades that are not coordinated between multiple Agencies and Programs. The first Level-3 processing step is to align these standards between all missions, and to ensure that SWOT products use the latest recommendations from the Ocean Surface Topography Science Team, and SWOT Science Team: all missions are upgraded consistently at any given time.



To illustrate, at the time of this writing, the Level-2 products version "C" from SWOT are not aligned with the DT-2024
standards from Kocha et al (2023) [1], although they will likely be upgraded in the near future. More generally, the details of
each upgrade might become obsolete as new corrections and models get integrated, and the input Level-2 is adapted to DT-
2024, so we will not detail minor upgrades (e.g. ancillary data such as the distance to coast, shoreline, sea-state-bias
implementation, etc.). However, there are at least three noteworthy items for the Level-3 product: the barotropic tide model,
the mean dynamic topography and the mean sea surface.

The DT24 standards selected the latest iteration of the FES22 barotropic tide model (Lyard et al., 2024; Carrere et al. 2023)
after a side-by-side comparison with other recent tides models. FES22 significantly reduces the measurement errors in many
coastal regions (Figure 4). The improvement measured with Jason-3 can be as large as 2 to 4 cm² for the coastal ocean, whereas
the TOPEX/Jason ground track was assumed to be well-charted for tides. Similarly, for higher latitudes not covered by Jason-
3, and particularly for the Arctic Ocean, the FES22 model was reported by Carrere et al. (2023) to reduce the variance by 2 to
5 cm² in the deep ocean and 20 cm² or more in coastal regions. Beyond the specific case of FES22, the barotropic and baroclinic
tide models are extremely important for SWOT because it is the first mission able to chart all coastal regions with 2D images
at 250 m or 2 km resolution and high precision. To that extent, the SWOT 2D images are a lot more affected by tide model
residual errors since they are observing new geographical zones, compared to the 1D repeat ground-tracks such as the TOPEX
to Sentinel-6, or ERS to SARAL, or Sentinel-3 satellites. In that context, it is essential for the Level-3 product to stay very
close to the published state-of-the-art, and to serve as an evaluation sandbox for any new tide models.

Similarly, the DT24 standards use the CLS22 mean dynamic topography (MDT) model from Jousset et al (2023). This model
brings a significant improvement upon the 2018 iteration, in particular over the Arctic Ocean. In addition to a more complete
coverage, the 2022 MDT model also fixes documented artefacts and improves the consistency with independent drifters by
10% on average, and locally often more.

The most essential upgrade for KaRIn is the mean sea surface (MSS) model. At the time of this writing, the Level-2 SWOT
products version "C" are based on the 2015 CNES/CLS model. In contrast, the Level-3 uses the so-called Hybrid 2023 that
blends the best of CNES/CLS22, DTU21, and Scripps22 (see Laloue et al. 2024; or Schaeffer et al., 2023 for details). This
model was shown to be significantly better than any alternative, especially for the smaller scales of interest with SWOT. Figure
5 illustrates this with an arbitrary example near the Dominican Republic. Panel (a) shows the SSHA from KaRIn: the zoom
from 20 to 22°N exhibits some very unusual features for SSH anomalies. They are explained by panel (b) and the GEBCO
bathymetry. The SSHA features are strongly correlated with the bathymetry: the strange topography features actually originate
from unresolved geoid signatures that the 2015 CNES/CLS MSS model does not correctly represent. This arbitrary example

---

[1] The details of DT2024 standards are recalled in the appendix of section 9.





is representative of the many residual geoid signatures that one might see in KaRIn images wherever the bathymetry is rugged: uncharted seamounts, rifts, continental shelf…

Furthermore, Figure 5c shows the power spectral density of the KaRIN SSHA based on two different MSS models (derived from Laloue et al, 2024): the MSS15 model from the Level-2 product and the MSS23H from the Level-3 product. While the MSS has essentially no impact on scales below 10 km or above 70 km, there is a critical range of wavelengths from 10 to 70 km, where the old MSS model from 2015 increases the variance by as much as 90%. Moreover, it tends to distort the PSD shape by adding a hump-shaped artifact: this artifact is fully explained by the MSS error (red and blue lines). These

observations are not surprising as they were predicted by Pujol et al (2018). The new hybrid MSS model is also beneficial for traditional 1D altimeters, but not as much: their measurements are increasingly contaminated by instrument noise below 70 km so the MSS errors are less impactful than for KaRIn. To illustrate, the thin black and grey lines from the two MSS models applied to the SWOT nadir altimeter differ by only 10% from 10 to 70 km. In contrast, having an accurate MSS model becomes crucial for the KaRIn SSHA because of the very high precision of this instrument.

Despite the improvement brought by the 2023 MSS model, there are still many regions where residual geoid features are very noticeable as in Figure 5a. To that extent, it remains extremely important to keep improving MSS models for the critical scales ranging from 10 to 70 km. In that context, by far the main asset for the long term is the KaRIn SSH measurement itself, as it can be averaged into a local mean sea surface or mean profile (Dibarboure and Pujol, 2021 ; Yu et al., 2024).

### 3.2 Editing spurious pixels

The Level-2 product contains a large number of quality flags, which provide a flexible way to remove spurious pixels. In addition to the Level-2 product documentation (section 10), Chen et al. (2023) explain in detail their meaning and they provide a general guideline of how to use them. While extremely useful, these flags are sometimes based on theoretical uncertainty and they do not always capture the reality of suspicious/spurious pixels. Some flags such as the sea-ice flag, radiometer rain flag, or KaRIn wave quality flags (among others) are sometimes detrimental: they tend to remove large segments of images,

which are perfectly fine. In contrast, none of the Level-2 flags properly captures obvious anomalies such as heavy rain cells, or some sea-ice. Moreover, the shoreline mask used to discriminate land and sea in the ground segment is not always reliable, resulting in uneven flagging of the shorelines and islands with detrimental effects in coastal zones.

In order to improve the spurious pixel and outlier detection, the Level-3 processing uses a more practical and data-driven editing sequence. More precisely, the L3 editing uses the following incremental steps of decreasing importance:

• Flag value 102: Suppress defaulted SSHA values (e.g. because of missing KaRIn SSH or geophysical corrections and models)





- Flag value 101: Suppress pixels over land using a shoreline mask. This step is similar to the land flag in the Level-2 product but based on a custom land/sea/island/lake mask combining different sources (e.g. Open Street Map, Global Island…).

- Flag value 100: Suppress pixels if the KaRIn Level-2 quality bit 256 is raised, but only for the edges of each swath. The edges are at the limit of the KaRIn coverage and they are not always properly covered at 250-m, resulting in various artefacts when the data is averaged down to 2-km. Although these columns are beyond the SWOT requirements (10 to 60 km), we do try to keep the edges when their coverage is good enough.

- Flag value 70: Suppress pixels impacted by spacecraft events (e.g. gyrometer calibration, maneuvers, eclipses
transitions...) based on the Level-2 quality bit 2048. Most of these events can affect the SSHA quality.

- Flag value 50: Suppress abnormally high SSHA values in coastal and Polar Regions, as they are generally caused by land or ice contamination in mixed pixels (i.e. iceberg in an ocean pixel, layover from land or sea-ice).

- Flag value 30: Suppress SSHA pixels that are out of the expected statistical distribution, i.e. a custom look-up table based on the KaRIn noise as a function of significant wave-height.

- Flag value 20: Suppress the KaRIn-based L2 quality bits $2^8=256$ and $2^{30}=1073741824$ in Polar Regions. This step removes suspected sea-ice pixels. Note that this editing layer tends to be very conservative, and to keep only open ocean pixels: users who want to study the polar ocean in the presence of sea-ice may want to ignore this editing layer in order to keep as much coverage as possible.

- Flag value 10: Suppress all pixels flagged in the Level-2 product if they are less than 5 km from the coast and flagged in
the Level-2 product (parameter *ssha_karin_2_qual* > $2^8$). In the last couple of coastal pixels, various anomalies may appear from KaRIn measurements, or from geophysical corrections (imperfect tides or atmospheric correction, imperfect mean sea surface models, etc.). To that extent, this editing layer tends to be conservative to keep only trustworthy pixels based on Level-2 flags: users who want to keep as much coastal coverage as possible may want to ignore this editing layer.

- Flag value 5: Suppress all pixels that deviate from a locally smoothed version of the KaRIn SSHA (Gaussian filter, cutoff
20 km). The purpose of this step is to identify smaller scale discrepancies such as ships, icebergs or tiny isolated rain cells that were not detected by previous steps. This last layer is not applied in a handful of regions with very large internal tide signatures (North of Brazil, Madagascar, Indonesia and Malaysia, Bay of Bengal). For these regions, it tends to be too conservative and to misinterpret the sharpest internal wave crests or troughs as spurious regions. We think it is better to leave some rain cells rather than altering actual content in a systematic way, at least until a better editing layer can be
developed.

To summarize, the editing flag falls into one of four categories:

- 0 means the pixel went through all the editing layers and is considered good;





- 10 and less means the pixel is suspect so it is recommended to ignore it if the user wants to make sure they keep only the best quality;

- 11 to 99 is a mixed bag of bad indicators of increasing importance, and it is recommended to ignore these pixels unless more coverage is really needed (e.g. coastal regions, see below);

- 100 and more means that it is strongly recommended to not use the pixel as it is very likely unusable or defaulted.

Figure 6 gives three examples of SSHA before editing (left column), and after editing (center column) as well as the corresponding editing flags (right column).

- Panel (a) is a typical example of flag values 100 or more (red regions) where we discard cross-track distances beyond the nominal 10 to 60 km. It is also an example of flag value 70 in an eclipse segment: in this particular example, the SSHA data is fine, even if the L2 product has raised the mission event flag (discussed below). Lastly, the spurious pixels from rain cells are captured by lower flag values (typically flag values 20 to 70, i.e. cyan pixels on the right panel).

- Panel (b) is an example of coastal region where intermediate flag values tend to remove the last coastal pixels while higher values remove the land pixels. In this particular region, the coastal SSHA is likely degraded by geophysical corrections, rather the KaRIn measurement itself. So, removing these pixels is a questionable and subjective choice. For many users, taking out degraded SSHA makes sense. However, the only way for some experts to retrieve better MSS or tides models with SWOT data would be to keep these pixels as an input to their studies. Consequently, tides

and MSS communities may prefer to ignore some of these quality flags.

- Panel (c) is an example of a region with massive internal tides: the challenge here is to isolate rain cells while not removing the smaller scales of internal tides. In this particular example, the editing is working well, but in other regions, it might be too aggressive and remove useful internal tides, or internal waves. To summarize, the current L3 editing layers does a good job to remove spurious pixels that are not always captured by Level-2 flags, but it may be

over-aggressive in ambiguous situations.

Moreover, Figure 7 shows two editing examples at 2-km (panels a, and c) and 250-m (panels b and d):

- The first example of panels (a) and (b) is in the Tropical Pacific Ocean, with a mix of mesoscale, internal waves and some rains cells, with some (expected) spurious pixels for the columns on the swath edges [2]. In this particular scene, the 2-km editing takes out the swath edges (red and orange pixels in the right-hand side editing flag map). Then the

bulk of the rain cell is isolated by intermediate editing layers (green pixels of the editing flag map). Finally, the last

---

[2] Most of these artifacts have two causes: the phase screen correction, which is not final in the L2 version 'C', or the 250-m to 2-km averaging process in a fixed geographical grid. The former exists at both resolutions, while the latter exists only in the 2-km resolution products, but not in the native 250-m pixels.





layers capture leftover pixels and rain cell boundaries (blue pixels of the editing flag map). The full editing flag tends to clean up most of this scene, but it is also somewhat aggressive as it destroyed a fraction of the internal wave feature. In contrast, the 250-m editing algorithm leverages the full resolution to isolate smaller clusters of spurious pixels, some of which would be averaged down to 2-km, and cause a small but significant degradation of the 2-km pixel that

300       is not captured by the 2-km editing.

- The second example is in a sea-ice covered region of the Southern Ocean. As expected, the 2-km resolution provides a coarse topography measurements where it is complex, if not impossible, to properly identify spurious pixels. Conversely, the 250-m editing retrieves a more consistent coverage with sharper detection of leads to floes transitions or ice/ice transitions. While this current editing algorithm was not designed to classify the surface type, this specific

305       example illustrates the capabilities of the 250-m resolution and the potential to finely separate the ocean from the sea-ice in the future.

Quantitatively, on average, 15 % of the KaRIn pixels are edited out if we include flags values 70 and 100. This value is also expected: the main contributor is the Level-2 quality bit 256, which is very often raised for cross-track distances beyond the 10 to 60 km requirements (i.e. approximately 8% of the time). Therefore, the KaRIn Level-3 algorithms flag approximately

7% of invalid pixels in the nominal 10-60km swath. That number includes the data flagged during the eclipse transitions (3 to 4%).

The eclipse transitions correspond to a small segment of 2 min along each orbit circle where the satellite transitions from an eclipse (in the Earth's shadow) to being illuminated by the Sun. During the eclipse transitions, KaRIn is no longer constrained by its requirements, as conservative pre-launch predictions assumed the instrument would need a couple of minutes to stabilize

to the new thermal conditions. However, recent analyses find that the data has a nominal quality during the eclipse transitions and that it should probably not be flagged. Unfortunately, eclipse transitions are currently flagged in the Level-2 product in an irreversible way: a single L2 quality bit is used to flag all mission events. In the current L3 release, we do not isolate the eclipse transitions from other events. Future releases of the L2 or L3 will probably retrieve the 3-4% percent of data that are currently edited out incorrectly during these eclipse events.

If we ignore the swath edges and eclipse transitions, which are excluded from the mission requirements, we have approximately the same fraction of edited pixels as nadir altimeters (3 to 4%). Moreover, the map from Figure 8a shows that this percentage increases in Polar Regions (expected from sea-ice), in rain regions as for SARAL and its Ka-band nadir altimeter (e.g. Picard et al., 2021), and in some coastal regions. For the latter, the fraction can be very large in some places (up to 50% for the last couple of coastal pixels): we suspect that geophysical corrections and incorrect MSS models are the primary sources of error,

at least up to the last pixel where KaRIn SSH artifacts start to dominate.

Still, the editing layers do meet their primary objective: Figure 8b shows the SSHA variance reduction when the editing flag is applied, versus when it is not. Dark blue region where the variance is reduced after editing are very consistent with tropical





rain regions observed by SARAL (Picard et al., 2021), or sea-ice regions. In these regions, the L3 editing reduces the variance by up to 50 cm²: this highlights it is essential to take out spurious pixels from KaRIn images, as they might induce artifacts

that are large enough to affect oceanography studies. However, the L3 editing also removes a large amount of variance in some coastal regions, which might be more questionable. Without a solid ground truth, or trustworthy validation metrics, we cannot determine if it is correctly removing artifacts from tides and MSS models, or if the editing also removes actual ocean features of interest as well.

To that extent, the editing layer is optional in the Level-3 product so that any expert user can decide to use it or not, or even

design their own editing strategy. In the 'basic' variant of the Level-3 product, all editing layers are applied by default, whereas the 'expert' variant has the flag itself and the unedited SSHA: the L3 user can therefore opt to keep only some editing layers if they want. This is particularly useful for lower values of the editing flag, as these layers can be somewhat subjective and we have no ground-truth to parameterize them with confidence. To illustrate, polar and coastal users, may prefer to suppress only the pixels with an editing flag of 70 or more. They will get more coverage, but also more suspicious pixels: this strategy would

also make sense for MSS or tides experts who want to get as much coverage as possible since they have their own theme-specific algorithms to suppress residual outliers.

### 3.3   Data-driven calibration

The images provided by KaRIn can be biased or skewed by a few centimeters to tens of centimeters (e.g. Figure 3a). There are various sources of errors: uncorrected satellite roll angle, interferometric phase biases, thermo-elastical distortions in the

instrument baseline and antennas, etc. In order to mitigate these topography distortions, it is necessary to use a calibration mechanism based on the interferometric phase or topography data itself, hence the name of data-driven calibration. Dibarboure et al. (2022) give an overview of the calibration of KaRIn images: why a calibration is needed, how it can be performed, and what was the expected performance before SWOT's launch. Moreover, Ubelmann et al (2024) provide a post-launch update: giving the magnitude of the systematic errors observed on real data, and the residual error once data-driven calibration is

applied. While the technical details are beyond the scope of this paper, this section intends to give a quick summary of the items of interest for the Level-3 expert users; standard ocean users can probably skip it and move on to section 3.4 (noise mitigation).

In both calibration papers, two variants of the calibration are described: the SWOT mono-mission or Level-2 algorithm, and the multi-mission or Level-3 algorithm. As the names imply, the former is used in the SWOT ground segment and L2 products,

whereas the latter is specific to Level-3 processors. The L2 algorithm was primarily designed to meet Hydrology requirements, and it is considered optional over ocean since it is not necessary to meet SWOT's ocean requirements from 15 to 1000 km. This algorithm is based on SWOT data only, because a ground segment cannot depend on external satellites.



In contrast, the Level-3 algorithm was designed to leverage better algorithms and external satellites: not only Sentinel-6, the so-called climate reference altimeter, but also all other altimeters in operations (Sentinel-3A/3B, HY2B/C, SARAL, CRYOSAT-2). The L3 correction is generally more robust and stable than the Level-2 variant thanks to the thousands of daily multi-mission crossover segments provided by the constellation.

$$SSHA_{cal}(t,b) = SSHA_{uncal}(t,b) + XCAL_{L3}(t,b) \qquad\qquad \text{Eq (1)}$$

$$\text{where } XCAL_{L3}(t,b) = B_{L3}(t) + b.L(t) + b^2.Q(t) + PS_{L3}(b,t) + SC(b) \qquad \text{Eq (2)}$$

$$\text{and } \quad B_{L3}(t) = B_{karin|nadir}(t) + B_{nadir|S6}(t) + B_{nadir|lwe}(t) \qquad\qquad \text{Eq (3)}$$

From a practical point of view (equation 1), the 'basic' variant of the Level-3 product contains a calibrated topography ($SSHA_{cal}$) whereas the 'expert' variant provides both the original/uncalibrated topography ($SSHA_{uncal}$) and the correction ($XCAL_{L3}$). The rationale is to give some flexibility to the users who want to use their own calibration instead. The calibration is provided per pixel, i.e. in 2D as a function of time $t$ (along-track coordinate) and cross-track distance $b$ (cross-track coordinate). The correction has five components (equation 2):

- $B_{L3}$ is a time-evolving bias per swath,
- $L$ is a time-evolving linear cross-track correction per swath,
- $Q$ is a time-evolving quadratic cross-track correction per swath,
- $PS_{L3}$ is our empirical estimate of the so-called phase screen: a small and slowly time-evolving bias that changes in the cross-track direction, see Peral et al (2024) for more details,
- $SC$ is the so-called static calibration of the B/L/Q parameters (mean value throughout the mission's lifetime).

At the time of this writing, the bulk of the $SC$ parameter is already accounted for in the Level-2 algorithms (e.g. meter-scale tilt of each image from the mean $L$, decimeter-scale curvature from the mean $Q$ parameter). The residual SC component we retrieve in the L3 is probably explained by the period where the mean was computed: the static calibration of the ground segment was computed in spring 2023, while our residual $SC$ is recomputed over a much longer time series and a consistent set of beta angles. As for the phase screen $PS_{L3}$, we retrieve an error of the order of ±4mm: most of the error is time-invariant, although a fraction of ±1mm continuously evolves along the orbit circle (the error changes with latitude, and it is different for ascending and descending passes). Note that in the future, it is likely that the $PS$ (phase screen) and $SC$ (static calibration) components will be zero, as they should be corrected in the reprocessed Level-1B algorithms.

Conversely, the primary purpose of the data-driven calibration is to handle the time-evolving errors B/L/Q, i.e. a second order polynomial function of $b$ for each swath for each time step $t$. Ubelmann et al (2024) confirm with flight data that the pre-launch strategy from Dibarboure et al. (2022) is relevant. Before the data-driven calibration, they observe various scales in the B/L/Q errors: very slow drifts of the order of a few months, cyclic beta angle variations of a few weeks, more rapid changes of the





order of a few days, harmonics of the orbital revolution period, and random higher-frequencies (e.g. noise from the gyrometer). The order of magnitude of each scale is consistent with prelaunch simulations.

Once calibrated, the B/L/Q errors are strongly reduced. This is typically visible in metrics such as Figure 9a: the variance of the measured KaRIn SSHA as a function of the cross-track distance $b$. This variance is the sum of three items: 1/ KaRIn's systematic errors (the L and Q term increase as a function of b), 2/ natural SSHA variability from the ocean (on average, it is the same for all cross-track distances), and 3/ other KaRIn errors (which are cross-track invariant). Before calibration (not shown), the variance increases away from the nadir position and it can be as large as 35cm² in the far range and regionally

much more. After the Level-2 calibration (green curve), the error is strongly reduced, but the cross-track dependence is still visible with a couple of cm² for the 21-day orbit (green, plain) and much more for the 1-day orbit (green dashed)[3]. In contrast, after the Level-3 calibration (blue curve), the cross-track dependence is almost removed, and there is no difference between the left and right swaths. In other words, the L3 calibration has strongly reduced the systematic error terms L and Q.

Figure 9b shows the geographical distribution of the standard deviation of the KaRIn SSHA. This map combines multiple 21-

day cycles and it does not exhibit any obvious regional anomaly: it is very homogeneous and consistent with its nadir altimetry counterpart (not shown), and the variance is explained by the actual ocean variability.

In order to quantify the residual error and its spectral breakdown, we used a variant of the Ubelmann et al. (2018) methodology. Using cross-spectra between parallel columns of the KaRIn images, we can form a 3D cube of cross-spectra and we can use 2D frequency slices to separate the signature from correlated errors in the nadir and KaRIn datasets, and specifically isolate

the systematic error terms B/L/Q. The result is shown as a power spectral density (PSD) in Figure 9c. The black line is the KaRIn SSHA power spectrum (before calibration) and the brown/orange are the systematic errors.

Before calibration (dark brown), the error is mainly described as a $K^{-2}$ power law from 15 to 40,000km (orbital revolution) then a flat value for longer time series (multiple revolutions, hourly to daily drifts). On top of this background line, there is a series of peaks located on the harmonics of the orbital revolution periods (thermo-elastical distortions). The dark brown PSD

is very similar to pre-launch simulated errors: no significant changes of the calibration algorithms were necessary with actual flight data.

The orange brown curve is the residual error after Level-2 calibration (crossover mono-mission algorithm) and the light brown curve is the residual error after Level-3 calibration (multi-mission). As expected, both calibrations reduce the systematic error for longer wavelengths, by a factor of 10 for the L2 and a factor of 50 for the L3. The L2 calibration starts to be significantly

effective near 5,000 km, whereas the L3 starts near 1000 km. For scales smaller than 1000 km, KaRIn is meeting its

---

[3] The large difference between the two orbits is discussed by Dibarboure et al. (2022) and beyond the scope of this paper. In essence, the 1-day phase has very few crossovers because the sparse geographical coverage, which in turn thresholds the Level-2 calibration, which is based on crossover overlaps. The Level-3 algorithm leverages multiple nadir altimeter missions, which makes it much more resilient to this SWOT crossover sparsity.





requirements (red line, from 5 to 1000 km) without any form of calibration. Integrating the PSD for all wavelengths, the error after L2 calibration is of the order of 2.5 cm, and after the L3 calibration, the residual error decreases to 1.5 cm RMS. Most of the variance is located at the larger scales, and they probably originate in leakage of the ocean variability (e.g. uncorrected biases and slopes from barotropic corrections that might be ambiguous with KaRIn errors).

The calibrated results are consistent with the prelaunch simulations from Dibarboure et al. (2022). This is not surprising because the *uncalibrated* errors are also consistent with prelaunch simulations, and the algorithms are unchanged. Ubelmann et al (2024) make a more in-depth analysis of the L2 & L3 calibration algorithms and their validation, including over land.

### 3.4   Noise mitigation

KaRIn's precision has improved over conventional radar altimetry by more than an order of magnitude (Fu et al., 2024), but
SSH derivatives quickly amplify the millimeter-scale noise to the point where it can become the primary limiting factor (Chelton et al., 2022). This is clearly visible in the upper panels from Figure 10 and Figure 11 in the Gulf Stream region. For this reason, Treboutte et al. (2023) have developed a noise-mitigation algorithm based on a convolutional neural network (CNN) that was specifically trained with SWOT simulated data in order to reduce random noise, while leaving most of the signal intact. In essence, the CNN is able to discriminate decorrelated random noise from geographically correlated ocean
features. In simulations, this strategy performs better than kernel filters, or the variational filter from Gómez-Navarro et al. (2020).

Nevertheless, the prelaunch CNN did not work as expected on SWOT's flight data for two reasons. Firstly, the noise floor of flight data is significantly better than in simulations, as well as more correlated at 2-km than the pure white noise they used in simulations. Secondly, KaRIn flight data has a series of outlier and correlated glitches as well as some ocean features that were
not in the prelaunch simulated training dataset: e.g. rain cells that are not edited, spurious pixels from the coast or sea-ice, strong non-linear internal waves, etc. The pre-launch CNN generally behaved poorly when it encountered such features it was not trained against.

So the same CCN (UNet variant) was retrained on a more realistic training dataset: less random noise than in prelaunch simulations, training on the eNATL60 model with tides (Ajayi et al., 2020; Brodeau et al., 2020) to better simulate the SWOT
observations. The re-training was performed with a random wave-modulated white noise at 250-m, with a Hamming filter to mimic the downscaling to 2-km that is performed in the ground segment. Lastly, in order to generate a training dataset as realistic as possible, we used the style transfer technique from Gatys et al (2016): we used the simulated SWOT product as the first input and real SWOT images to define the target style. This approach yields a training dataset that is more realistic, with more diverse oceanic and atmospheric conditions. This retraining improved the denoising behavior in most places.

Some denoising examples are shown in Figure 10. The upper panels are the SSHA and some derivatives from the original SWOT images, and the bottom plots are the same scene after denoising by the UNet. In this Gulf Stream region, the UNet is





able to take out most of the noise on geostrophic velocities, without a massive smoothing of oceanic features as small as 10 to 20 km. Moreover, it seems to retrieve some vorticity features. Although the latter remains quite noisy, the UNet does yield a massive improvement upon the original vorticity (upper right panel). In this example, a traditional kernel smoother would
essentially smooth out most if not all small-scale features of second derivatives (Chelton et al, 2022).

Figure 11 is the same scene as Figure 10 but with the 250-m product. In this example, the geostrophic velocities are computed on a 9-point stencil like for the 2-km counterpart. This results in more noisy gradients since they are effectively computed with a distance 8 times shorter than in Figure 10. Still, the bottom-right panel of Figure 11 shows that the 250-m product can retrieve very small scales features with sharper gradients than at 2-km. This might prove beneficial to some specific users such as
geodesists that want to leverage the SSH slope to derive gravity anomalies and bathymetry.

On average, the UNet removes approximately 5 mm RMS from the SSHA. These values were found to be stable over a few months of the science orbit. KaRIn's noise floor is arguably closer to a couple of mm RMS, so the filter is probably removing more than just the random noise from KaRIn: it takes out either some ocean features or some correlated errors from KaRIn. This is confirmed by the PSD of Figure 12a. Before denoising (blue curve), the SSHA spectrum exhibits a spectral slope
transition from $K^{-3.5}$ (or $K^{-11/3}$) for the larger mesoscale to $K^{-2}$ below 70 km. This slope break might originate from unbalanced motions (internal tides, internal waves) or from red-colored KaRIn measurement errors (e.g. from atmospheric or sea-state sources). In contrast, after the L3 denoising (orange curve) the SSHA spectrum is linear from the larger scales to 15 km. In other words, the denoising seem to mitigate not just random white noise, but also a fraction of the $K^{-2}$ slope break below 70 km. The green curve is the PSD of the UNet residual and it is more or less following a $K^{-0.5}$ slope. In other words, the UNet
might be capturing either red-colored KaRIn errors (e.g. wet troposphere, atmosphere or rain or sea-state residuals) or ocean features in addition to instrumental random noise.

Furthermore, Figure 12b shows the spectra of similar plots for 24-hour differences[4] during the 1-day phase of SWOT. Before denoising (blue curve), a large fraction of the large mesoscale cancels out in 24h (as expected). In contrast, the linear part below 100 km is barely removed by the 24-h difference, i.e. the $K^{-2}$ slope discussed above is mostly decorrelated in 24-h,
especially at the smallest scales. After denoising (orange curve), the 24h differences are mostly the same for the larger scales but the linear trend is removed: the 24-h difference PSD (orange, plain line) slowly converges towards the PSD of the denoised SSHA. This is what one would expect from a PSD dominated by ocean mesoscale features, where the shorter wavelengths also decorrelate faster in time. The spectrum of the 24-hour difference of the UNet residual (green, plain line) is also very close to the mean PSD of this residual (green, dashed line), which indicates that the content removed by the UNet is decorrelated in
24-h (as one would expect from measurement errors, or atmospheric and sea-state effects, or internal waves). But the two

---

[4] A factor ½ is applied on the PSD of 24-h differences. The rationale is that if a given random signal S is decorrelated in 24h, the PSD (or variance) of the 24h hour difference $S_d$-$S_{d+1}$ will be equal to twice the PSD (or variance) of S because the co-spectrum (or covariance) is zero. The factor ½ will then re-align the PSD of $S_d$-$S_{d+1}$ with the PSD of $S_d$ and $S_{d+1}$.



green curves are not perfectly aligned, and there is approximately 30% of variance missing: either the UNet has captured some temporally coherent errors (unlikely for atmospheric or sea-state or random errors), or it has absorbed a small fraction of temporally coherent ocean features (e.g. higher modes of internal tides, a small fraction of the stationary mesoscale…).

Note that we do not have any ground truth to quantify the denoising performance. So it is difficult to determine if the noise
mitigation has only positive effects, or if the smaller features in the SSHA derivatives are also made of correlated noise. From the day-to-day consistency that we observe in subsequent images of the CalVal phase, we think there is desirable noise mitigation skill in the current combination. However, that statement is subjective and by no means a proper demonstration. The spectra of Figure 12 are consistent with what one might expect from a denoising, but we cannot rule out that the UNet is inventing or distorting some features to replicate the properties of its training dataset.

To that extent, the noise mitigated SSHA is provided as an optional asset of the Level-3 products: in the 'basic' L3 variant, it is provided in addition to the original raw/unfiltered SSHA. In the 'expert' variant, it is provided as a separate optional layer that can be ignored.

## 3.5    Topography derivatives

The Level-3 products from KaRIn also include geostrophic velocities and relative vorticity in 2D. This is a relatively simple
addition provided only for the sake of user convenience. As explained in the previous section, the derivatives are computed based on the noise-mitigated SSHA only. The differentiation is computed with a 9-point stencil, following the methodology of Arbic et al. (2012). The rationale is to minimize velocity-dependent biases. The stencils were originally developed for very coarse Level-4 altimetry maps (also known as AVISO maps), but Arbic et al. (2012) show that a 5 km resolution model still benefits from larger stencils.

## 4    Level-3 Products validation and use cases

The SWOT interferometer is the first instrument of its kind, its validation is still ongoing. At the time of writing, there is no public dataset able to provide a solid ground truth to evaluate the benefits from the Level-3 algorithms presented above. In previous sections, we illustrated and quantified how each layer of Level-3 algorithms brought a small improvement to the overall product. In this section, we analyze the complete Level-3 product.

In section 4.1, we provide a global overview of the product content, as well as some qualitative internal and multi-sensor comparisons. Then in section 4.2, we quantify the differences between SWOT and Sentinel-3 (operational altimeter from the European Program Copernicus) for Level-2 and Level-3 products.





### 4.1 Qualitative assessment

From a qualitative point of view, Figure 13 shows a gallery of examples of the Level-3 product from KaRIn. The global map
is a composite of 1 cycle of a 21-day orbit in November 2023. The subpanels below are a series of regional zooms over different
recurring features that may be seen in KaRIn images. For the global ocean, the SSHA is strikingly consistent with gridded
Level-4 products from the Copernicus Marine Service (sometimes referred to as AVISO maps), although for SWOT KaRIn
no interpolation is needed. KaRIn alone is able to cover most of the ocean a couple of times every 21 days (ascending &
descending passes).

Panels (a) and (e) show that the large scale of the Level-3 product is very consistent once it is calibrated: the former shows the
expected Indian Dipole in a positive phase, while the latter shows the propagation of the El-Niño event of Fall 2023. Even at
these larger scales, the Level-3 product is better than the Level-2 because of the newer tide model and the multi-mission data-
driven calibration.

Incidentally, panel (d) exhibits a series of vertical stripes that can be found in various regions. This pattern is expected from
such SWOT composites: it originates in the stroboscopic sampling pattern of the SWOT orbit. Indeed, the global coverage of
SWOT's 21-day orbit is assembled from 2 sub-cycles of 10.5-days each (Lamy et al, 2014). Each sub-cycle of 10.5 days
corresponds to a global coverage pattern that is interleaved (in longitude) with the coverage of the previous and following sub-
cycles. In other words, with every half sub-cycle, SWOT covers half of the ocean, alternating between two interleaved scan
patterns. Furthermore, the El Niño event is quickly propagating eastwards over a 10 day period, and the stripes indicate the
SSHA is rapidly evolving between subsequent interleaved scans from SWOT. For the same reasons, stripe-shaped
discrepancies can be found between adjoining swaths in all regions where the ocean changes significantly faster than the 10.5-
days it takes to get the interleaved scan.

Panel (g) shows a clear and consistent view of mesoscale eddies in the Southern Ocean. The larger mesoscale (150 km or
more) is relatively well-known from nadir altimetry Level-4 maps, i.e. after an interpolation of many 1D profiles spanning
over tens of days. In contrast, the Level-3 product from SWOT provides a synoptic view of the larger mesoscale without any
interpolation whatsoever: it captures the actual shape and amplitude of each eddy without the distortion and smoothing that
one might expect from the gridding procedure of Level-4 products. Furthermore, in addition to the larger mesoscale, medium
to small to submesoscale is also retrieved in Level-3 products. This panel contains some eddies as small as a 20 to 50 km.
These features are beyond the resolving capabilities of 1D altimetry (Ballarotta et al, 2019) for two reasons: 1/ the constellation
of 4-7 nadir altimeters does not have enough coverage to resolve these features in space and time and 2/ the precision of 1D
altimeters is, on average, insufficient to consistently observe them in 1D segments. To that extent, KaRIn is major breakthrough
in the observation of the smaller ocean turbulence (Fu et al, 2024). Note that these eddies near Drake Passage in Figure 13g
appear quite consistent between neighbouring swaths. This highlights a remarkable feature of the SWOT sub-cycles: the
ascending passes shown here from one 10.5-day subcycle are laid down over consecutive days towards the west. At mid to





high latitudes as the swaths converge, this allows a consistent composite 2D local sampling 600 km wide over 5 days, with only one day offsets between neighbouring tracks. Then the SWOT descending passes form another 5-day local 600 km wide sampling, then its repeated in the second 10.5 day subcycle. Figure 14 shows the same for the Northern Atlantic Ocean and the Gulf Stream region but now over two 10.5 day subcycles. The largest eddies are clearly visible in the SSHA and geostrophic velocities and are generally quite consistent over neighbouring passes: any discrepancy between adjoining passes clearly

outlines how fast eddies are moving in a few days.

Figure 15 shows a comparison between SWOT's geostrophic velocities (black streamlines) and an Ocean Colour image background for a cloudless day. In this example, the largest eddies (more than 150 km) are very consistent as expected, and the smaller ones illustrate quite well that SWOT-derived velocities provide a synoptic view of the dominating factor for the Ocean Colour advection in this region. This is particularly interesting for thin fronts and eddies of a few tens of kilometer.

While both variables are not expected to be fully consistent at all times, this comparison shows a good qualitative correlation: our Level-3 product is a good asset for multi-sensor comparisons. At these scales, the Level-3 product is better than the Level-2 because of the MSS and MDT models, the calibrated phase screen, and a better editing of spurious pixels.

However, there are two noteworthy features of the geostrophic velocities in Figure 14b that deserve a more thorough analysis: 1/ there are many filament-like features in the open ocean (much less over the continental shelf), that could be due to a

background correlated noise, and 2/ in the South-Eastern part of the maps from Figure 14, some stripe-shaped SSHA features of panel (a) are interpreted as geostrophic velocities in panel (b).

These patterns, also clearly visible in Figure 13h in the same region (or Figure 13b in the Luzon Strait), are actually caused by internal tides and not mesoscale dynamics. Indeed, KaRIn gives a single synoptic view of the scene, and although a coherent internal tide model correction has been applied for the SSHA, this model is not perfect and cannot correct for incoherent, non-

phase-locked internal tides or solitons. As a result, the SSHA still contains the signature of unbalanced motions and waves. When these waves are close to their generation sites, they are coherent and structured, and therefore clearly visible. However, once they interact with mesoscale of the same size and amplitude, it might become a lot more difficult to separate balanced and unbalanced motions from a single Level-3 product. This topic is quite complex and beyond the scope of this paper, but Figure 13 and Figure 14 illustrate how complex is the quantitative validation of SSHA without a global ground truth, and that

care must be taken in interpreting geostrophic velocities, since many SSHA features captured by SWOT are not in geostrophic balance.

A different approach to verify the qualitative consistency of the Level-3 product is shown in Figure 16. The KaRIn 2D image and the SWOT nadir 1D profile are shown on top the Level-4 multi-mission gridded altimetry product from the Copernicus Marine Service (here shown in its natural resolution of 0.25°, without graphical smoothing). The left and center panels are the

same regions, separated by 4 days, and the right panels are the difference. The top panels are in the Southern Ocean; in a region where mesoscale eddies barely moved over this 4-day lag (confirmed by the pixelated background from nadir altimetry). The





bottom panels are in the Kuroshio region, and they capture a very rapid transformation of the eddy field (confirmed by nadir altimetry, albeit very smoothed in space and in time). These cases are interesting for two reasons.

Firstly, they show the self-consistency of SWOT over subsequent days and different sea-state conditions. Indeed, panel (a)
and (b) are very consistent and the difference is small in comparison, despite major changes in atmosphere and sea-state conditions (more than 5 m) in four days. More importantly, the RMS of the KaRIn/KaRIn differences in panel (c) exhibit a well-behaved PDF (not shown) and a variance of 1 cm², i.e. much less than the nadir/nadir differences (3 cm²). The Southern Ocean is an interesting region for this qualitative validation exercise; especially over slow-moving or bathymetry-trapped eddies: their stability is a good way to confirm the repeating nature or KaRIn SSHA in different atmosphere and/or sea-state
conditions.

Secondly, Figure 16 shows the good consistency between SWOT Level-3 products (both sensors) and the nadir altimetry constellation, although the temporal smoothing of the 2D maps can be limiting its ability to serve as a ground truth for KaRIn validation. The eddies from panels (d) to (f), are moving very fast and the 1D altimeter constellation underestimates their deformation and actual shape. However, it does provide a trustworthy albeit blurry view of the mesoscale field to help
determine if the changes in KaRIn images are happening or not.

In that specific example, SWOT observes a consistent and rapid change: not only are the bigger eddies traveling faster than the interpolated Level-4 map might indicate, but their shape is anisotropic and evolving very quickly as the 4 eddies in the SWOT scene interact with one another. Panel (f) also shows that, in 4 days, the SSHA changes by 30 cm or more at the heart of the interaction. This is significantly more intense and faster than what is usually observed by Level-4 products from nadir
altimetry because the temporal resolution of this product is more than two weeks (Ballarotta et al, 2019).

More importantly for the Level-3 validation, this example from the 1-day phase captures the temporal evolution of the KaRIn product that seems intuitive and reasonably robust. We do lose small image fragments due to rain in panel (e) and (f) and some artifacts may remain, but the Level-3 captures the temporal evolution of the mesoscale during the 1-day phase.

Lastly, Figure 13 also illustrate some limits of the SWOT/KaRIn Level-3 product. In panel (c), many parts of the images are
missing. This is caused by tropical rain events that are edited out when the SSHA is not usable (see section 3.2). In panel (f), some swaths have a lot of visible noise. This is caused by the presence of a major storm and high waves (typically 6 meter or more): in these conditions, KaRIn images become a lot noisier even at this 2-km resolution (Peral et al, 2024), albeit much less than the profiles of traditional altimeters. Note that the noisy image segments remain usable anyway: the coverage remains good and the larger scales is not significantly degraded.

**4.2    Comparison with external altimeters**

In order to quantify the residual errors of our KaRIn product, we computed all crossover sections with altimeter profiles from Sentinel-3A and 3B, with a time difference less than 24 hours. When the solar time of SWOT is aligned with the sun-





synchronous orbit of Sentinel-3 (S3), there are thousands of crossover match-ups where S3 passes through the KaRIn swath, thus creating millions of pixel associations between both sensors. Moreover, when the S3 nadir profiles crosses the SWOT nadir profile, we also get nadir/nadir altimeter crossover points. More importantly, because SWOT does not use a sun-synchronous orbit, the crossover match-ups will migrate in space and cover all regions in a semi-systematic pattern. This provides a homogeneous coverage of the global ocean. Over these match-ups, we can use the SSHA difference between SWOT and S3 to infer how much error might be affecting our Level-3 product.

This is not trivial because the time difference between both measurements is not zero, and because Sentinel-3 has its own error sources. Dibarboure and Morrow (2016) performed a similar exercise with the geodetic phase of Jason-1, when the longitude-drifting tracks of Jason-1 align with the tracks of Jason-2. They used these long alignments of nadir tracks to quantify the variance, geographical distribution, as well as power spectra of the SSHA difference for 24-hours or less. To illustrate, they find a variance of the order of 3 cm RMS for 0-day difference (essentially the random error of both altimeters: 2.7 cm RMS each), which increases rapidly to 4 cm RMS if the time difference is 24-h or less (i.e. +3 cm RMS from the 24-h ocean variability).

In that context, Figure 17 shows some statistics on the SWOT/S3 differences: probability distribution function on the left and variance of the difference as a function of the cross-track distance on the right. The upper panels are for the Level-2 products of Sentinel-3, SWOT/Nadir and SWOT/KaRIn. This plot measures the discrepancies between Level-2 products before any Level-3 algorithm is used. This configuration yields the sum of SWOT and Sentinel-3 errors. The bottom panels of Figure 17 are for Level-3 products.

In panel (a), the PDF is well behaved for all sensors[5]. The mean of the SWOT/S3 difference is almost but not exactly zero: S3B crossovers exhibits a small bias of the order of 6 or 7 mm and KaRIn crossovers a bias of 1 or 2 mm. The standard deviation (STD) of these 24-hour crossover differences is approximately 5.4 cm for both SWOT instruments. In other words, the nadir altimeter or SWOT and KaRIn exhibit similar errors, and SWOT has slightly less bias than Sentinel-3B.

If we use the results from Dibarboure and Morrow (2016) to account for the known variance of Sentinel-3 errors and 24-h ocean variability, we obtain a residual of the order of 3.3 cm RMS for both sensors of SWOT (unexplained variance that might be interpreted as SWOT error). For the nadir altimeter, this number is consistent with its theoretical error budget (all SSHA components, including residual errors from geophysical corrections & models), so we can assume this metric is also a good ballpark estimate of the total KaRIn error in the Level-2 products.

---

[5] The KaRIn/nadir PDF is smoother than the nadir/nadir one. This is because each KaRIn/nadir crossover segments yields 50-200 times more pixel match-ups than nadir/nadir crossover points. The KaRIn/nadir crossover segments can be as long as thousands of the kilometer long when the SWOT/S3 solar times are aligned.



Discussing the details of each component in the SSHA error budget is beyond the scope of this paper, as their total error budget includes many components: precise orbit determination error, random noise, wet and dry troposphere, ionosphere and sea-state bias, as well as residuals from the data-driven calibration... Yet, some of these contributors are the same for both sensors (e.g. orbit determination, or geophysical models) while others are not. The two main differences are the long wavelength errors and random noise [6].

Indeed, KaRIn has much less random noise than a nadir altimeter, but it has additional sources of error that do not exist in nadir altimetry. In particular, the so-called systematic errors (e.g. uncorrected satellite roll, interferometric phase bias...) are mitigated by the data-driven calibration (discussed in section 3.3). This error source is visible in panel Figure 17b where the variance increases as function of the cross-track distance. The quadratic shape in variance is typically expected from residual satellite roll, or from actual ocean slopes that might have been distorted by our calibration. The error variance that increases

in the cross-track direction is mostly from the calibration residuals (+2.2 cm RMS), and the rest is from all other sources of KaRIn error (2.5 cm RMS).

Then we repeat the process with Level-3 products of SWOT and Sentinel-3 in the bottom panels of Figure 17. It is clear from panel (c) that the L3 calibration has significantly improved the consistency with Sentinel-3 for both SWOT instruments. For nadir altimeters, multi-mission calibration is known to reduce POD error, as well as residuals from barotropic corrections and

instrument & processing biases (Dibarboure et al., 2011). The benefits are the same for KaRIn, since the Level-2 sources of errors are the same. In addition, the L3 data-driven calibration better mitigates the KaRIn-specific systematic errors (e.g. spacecraft roll) thanks to multi-mission crossovers.

The resulting PDF of the SSHA crossover difference is still well behaved, there is a bias of only a couple of millimeters, and a STD of 3.9 cm for SWOT/nadir, and 3.7 cm for KaRIn. Contrary to the errors observed on L2 products, here KaRIn exhibits

less error than the SWOT nadir altimeter (-1.2 cm RMS): this is because the L3 algorithm reduced POD or tide/DAC residuals for both instruments, and the L3 calibration also removed a bigger fraction of KaRIn's systematic error. Furthermore, Figure 17d barely exhibits any quadratic shape in the cross-track direction: the error is slightly higher on the edges of each swaths, by less than 1 cm RMS, which indicates that the L and Q components have been reduced to a residual that is barely measurable. If we account for the S3 error budget and natural variability in the SWOT/S3 24-h difference, that leaves approximately 1.8

cm RMS of other errors in the L3 product (e.g. POD, geophysical corrections incl. tides & atmospheric correction, noise, biases of all sorts).

---

[6] There is also a third difference in the ionosphere error. KaRIn is in Ka-band and therefore seven times less sensitive to ionosphere errors, while the SWOT and S3 nadir altimeters are un Ku/C bands and use a filtered dual frequency ionosphere correction. As a result, the ionosphere is a small contributor to the S3/SWOT differences. We will make the approximation that it can be ignored.





To summarize, the sum of all KaRIn Level-2 errors (including calibration residuals and geophysical corrections) are of the same order of magnitude as the total error of the nadir altimeter Level-2. However, the nature of this error is different: the nadir altimeter has more random noise, while the KaRIn interferometer is more affected by long-wavelength calibration
residuals. In the L3 products, the total error of both instruments is reduced by the multi-mission calibration scheme. And because the KaRIn total error is dominated by L2 calibration residuals, that are better calibrated with the L3 multi-mission algorithm, the error of KaRIn L3 product gets smaller than the nadir counterpart.

Furthermore, Figure 18 gives some insights on the geographical distribution of the error:

- The left panels (a) to (c) show the mean of the KaRIn/S3 difference
- The right panels (d) to (f) show the variance of the KaRIn/S3 difference.
- The upper panels are based on Level-2 products for both missions. This plot measures the discrepancies between Level-2 products before any Level-3 algorithm is used. The two datasets are completely independent.
- The center panels are based on Level-2 products for SWOT and Level-3 products for S3. At this point SWOT and S3 are still completely independent. Therefore, if the consistency with SWOT Level-2 is improved, it means the nadir
L3 calibration of Sentinel-3 reduced some long-wavelengths errors that are specific to this mission (as expected).
- Lastly, the bottom panels are based on Level-3 products for both missions. Here, both S3 and SWOT are ingested by the SWOT L3 processor, and the two missions are no longer independent. By construction, the L3 process reduces the discrepancies between SWOT, S3 and the rest of the constellation. These panels do not capture the SWOT error since both missions might have common errors (e.g. inherited from the reference altimeter Sentinel-6). Nevertheless,
these panels gauge whether SWOT is properly blended in the altimeter constellation.

Although the global bias between SWOT and S3 was very small in Figure 17, regional biases in Figure 18a can be as large as a few centimeters in the Eastern Pacific Ocean and the Indian Ocean. Such a dipole could originate in small POD residuals. Moreover, Figure 18b shows that calibrating S3 with the L3 multi-mission processing (without KaRIn) tends to reduce the regional bias with KaRIn. In other words, the bulk of regional biases from panel (a) are likely caused by S3 rather than SWOT.
Lastly, as expected from the multi-mission approach, when the Level-3 calibration is applied to both S3 and KaRIn in Figure 18c, the bias is negligible almost everywhere.

Similarly, the maps Figure 18d to Figure 18f show where the variance of the SWOT/S3 difference is located. Panel (d) is for the Level-2 products: the variance is rather homogeneous in the open ocean and slightly higher over the continental shelf, and higher latitudes and some latitude bands. In panel (e), the calibration S3 substantially reduced the error in many regions, i.e. a
fraction of the error of panel (a) is actually from Sentinel-3. Lastly, for the Level-3 in panel (f), the variance is no longer homogeneously distributed. The background variance in the open ocean is reduced (by L3 calibration). The continental shelves have much less variance than in panel (d). Firstly, the improved geophysical corrections (see sections 3.1 and 9) mitigated the SWOT/S3 standard discrepancy. Secondly, the L3 calibration absorbed a fraction of residual errors from barotropic tides





corrections or atmospheric corrections. Moreover, the spurious red pixels of this map mostly disappear (better L3 editing
process from section 3.2). The remaining variance in L3 crossovers is correlated with the geographical distribution of tropical
rain and wet troposphere, as well as storm tracks, or internal tides. There is also a thin alignment near 65°S with might be
related to the flagging process of sea-ice for SWOT. The residual discrepancies between S3 and KaRIn L3 products can be
explained by suboptimal geophysical corrections on SWOT, or Sentinel-3 or both.

To summarize, KaRIn / S3 crossovers illustrate the good consistency between both missions with Level-2 products, and the
excellent blending after Level-3 algorithms. On average, KaRIn has slightly less errors than SWOT's nadir altimeter in the
Level-3, because of editing, geophysical correction, and data-driven calibration. Still, both sensors from SWOT exhibit an
error level of the same order of magnitude, with more random noise for the nadir instrument, and more long wavelengths error
for KaRIn (of the order of 2-cm for the L2 and 1 to 1.5 cm for the L3).

## 5    Discussion: relevance and limits of this dataset

In this section, we extend the examples from Figure 1 and Figure 13: we try to illustrate the practical utility of our SWOT
Level-3 product for different research domains. We also try to discuss some limits of this dataset, and how collaborative
improvements with experts of various fields, and with the SWOT Project, might yield better products in the future.

### 5.1    Global circulation and Ocean Mesoscale

From the previous sections, it is clear that KaRIn has unprecedented capabilities to observe large and small mesoscales. The
gain in resolution is clearly visible in the Gulf Stream figures of Figure 1, and the ability to explore rapid changes in the
mesoscale field is shown in Figure 16. Moreover, Figure 13 illustrates quite well how the large and small scales are properly
retrieved, and how the orbit configurations allow a synoptic view of KaRIn, ranging from large scale basin-wide events such
as El-Niño or the oscillations of the Indian Ocean Dipole, to the anisotropic structure of small mesoscale eddies.

This paper has concentrated on the L3 processing of SSHA, and its data quality and validation. However, SWOT also has a
rich assortment of other 2D observations and derived geophysical parameters that are collocated with L3 SSHA. One key 2D
observation is sigma-0, calculated from the SAR processing of the 2D swath images, and is related to surface roughness and
wind speed algorithms in altimetry applications. Figure 19 illustrates that, under low wind and wave conditions, it is possible
to push KaRIn largely beyond its prelaunch requirements (SWOT, 2024). In this figure from the Mediterranean Sea in autumn,
panel (a) shows an extremely precise sigma0 from the Level-2 product at 250-m resolution, and panel (b) is our Level-3 SSHA
(250-m resolution) with black streamlines from the 2-km geostrophic velocities. In these low wind and wave conditions, panel
(a) reveals a wealth of thin oceanic filaments highlighting the interactions between the surface roughness and the small to sub-
mesoscale turbulence, i.e. the KaRIn counterpart of biogenic oil films seen on SAR imagers (e.g. Nichol, 2023) and/or the
processes described by Rascle et al. (2014).




Although some similar filaments of a small amplitude can be seen in various regions of panel (b), these features are generally
715 (but not always) averaged out at 2-km. The SSHA generally represents mesoscales that are stirring the ocean from depth. In
these calm conditions, the SAR sea surface structure is being partially set by the deeper mesoscale stirring, but the SAR image
is capable of detecting the near-surface layer that clearly evolves with much smaller features. The cause of these thin
topography filaments and their correlation with sigma0 images has not been fully described in the literature. They might be
real or artifacts. This effect might also exist in nadir altimeters, but hidden by their much larger noise envelope, or simply
720 harder to detect in 1D profiles. This example is one of many observed from SWOT, that would clearly benefit from in-depth
investigations from submesoscale and sea-state experts, and remote sensing experts.

For global ocean circulation and mesoscale studies, the most useful data from both instruments of SWOT is built in the Level-
2 products from the SWOT Project. Our additions in the Level-3 layers help reduce local biases (e.g. with state-of-the-art
geophysical corrections, or multi-mission calibration), or to better remove artifacts that might otherwise alter mesoscale
statistics, SSHA derivatives and their interpretation (e.g. editing or noise mitigation). For smaller scales and surface
interactions, the 250-m Level-3 product ("unsmoothed" variant) assembles all the assets needed to explore the 250-m
resolution conveniently (in comparison, the Level-2 unsmoothed product is technical but requires more specialized
processing).

### 5.2 Tides

Fu et al (2024) reports how KaRIn's resolution and precision is a major breakthrough in the analysis of barotropic and
baroclinic tides (internal tides or IT). The barotropic part is relatively known in the open ocean where recent tide models such
as FES or GOT are as precise as 5 mm RMS. In contrast, the coverage of nadir altimeter is sparse and insufficient in regions
of shallow bathymetry, in coastal seas, or in the high-latitude polar regions. Using KaRIn 250-m or 2-km products to improve
barotropic tides requires a good SSHA accuracy over scales ranging from a few tens of kilometers in coastal regions to
thousands of kilometers for the continental shelf to the open ocean.

In that context, an important limiting factor is the KaRIn calibration process. The uncalibrated KaRIn images (see Figure 3)
can be skewed by tens of centimeters (residual error of the satellite attitude determination) across or along-track that may be
at similar scales to barotropic tides, so it is necessary to use a data-driven calibration mechanism such as our Level-3 algorithm
from section 3.3 or the Level-2 counterpart. Since these calibration algorithms are data-driven (i.e. empirical), they may also
absorb some geophysical residuals as well (e.g. residuals after the FES22 model has been applied). This might be good for
mesoscale users, since tides are generally considered "an error" by this community. However, tides residuals are a signal for
tides expert. To that extent, the L2/L3 calibration process may be partially detrimental for this community.

As the tide community starts analyzing longer time series of SWOT data, more in-depth analyses of the Level-2 and Level-3
calibrations is required in collaboration with them, to quantify how this geophysical content might leak into the data-driven





calibration corrections. This future collaborative analysis could lead to better calibration algorithms over the ocean, and improved tide models in the key coastal, regional and high-latitude regions.

The baroclinic tides have very different paradigm. Although SWOT products have a baroclinic tide model correction applied to reduce their signature in the SWOT SSHA, this model only captures the first mode (i.e. larger scales) of a few constituents, as well as only the stationary part of internal tides. Figure 20 illustrates how KaRIn L3 data capture a vast range of baroclinic

tide signatures that are not corrected by these models. Panel (a) is a segment of SWOT measurements (altimeter and interferometer) in the North Atlantic. In the southern part of this region (right panel), there is smooth bathymetry where the internal tides look like very regular wave trains every 12h. Conversely, in the northern part of the scene (left panel), the bathymetry is rugged (mid-Atlantic rift), and the nature of the baroclinic internal tide (IT) surface topography changes: the KaRIn image looks like scrambled or random wave patterns.

Although the mode 1 and 2 (i.e. more than 50 km) are present in both scenes, the striking features of KaRIn's observation are the smallest scales and non-linear fraction of the waves: they aggregate into thin lines of the order of 10 km or less. This is the size of the SWOT nadir altimeter footprint. The SWOT nadir altimeter also has more random noise than KaRIn does. As a result, the altimeter does provide a consistent measurement of the internal tides in Figure 20a, albeit a smoothed and noisy one. For this reason, current internal tide models, which were built from 1D nadir altimetry, do not capture the full extent of

the internal tide signal nor their spatial structure. The residual is visible in all KaRIn SSHA products, including our Level-3.

Preliminary analyses of KaRIn-observed internal tides report that a substantial part of their variance is non-stationary, so current IT models only capture a relatively small fraction of the variance that is observed by SWOT. However, the KaRIn breakthrough of 2D observation for baroclinic tides should lead to progress in their modeling and their "correction" in altimetry (1D & 2D).

Our current Level-3 processing might be affected by these IT residuals at different steps of the processing. The editing and noise-mitigation might be confused by the smallest ripples of the baroclinic tides SSHA signature, and try to flag/remove them. Moreover, when the propagation of internal waves is orthogonal to the satellite track (e.g. circular wave of the right-hand side zoom of Figure 20a above a seamount), the apparent along-track wavelength might be long enough to leak into the data-driven calibration. This phenomenon was observed in the Bay of Bengal (not shown) where IT wave trains radiate from the eastern

part of the Basin (Andaman Islands): the SSHA exhibits very coherent wave signatures over more than 1000 km that are aliased at tidal frequencies. Similarly, in the Mascarene Region (Figure 20b), circular patterns are visible and geographically coherent over very long distances, and are shifted but visible on neighboring tracks separated by some days. It is possible that a small fraction of this IT variance leaks into our data-driven calibration.

These findings have two consequences. Firstly, it could be important for baroclinic tide experts to explore how the Level-2

and Level-3 processing affect their signal of interest when they use SWOT measurement to develop new and better models. Secondly, it is clear that progress is needed on the IT model "corrections" for KaRIn, and SWOT is an excellent testbed for




new models. More generally, the separation of balanced and unbalanced motions is extremely important to better interpret KaRIn images. As discussed in section 3.1, our Level-3 processor can integrate state-of-the-art research-grade models for large-scale evaluation of these models beyond the expert tide community. In other words, our Level-3 framework is a good

example of a sandbox product for testing new candidate tide algorithms and models, before they are adopted by the community into the SWOT Level-2 ground segment.

### 5.3    High-frequency atmospheric effects

Like all nadir altimeters, SWOT products are corrected for various atmospheric effects: radar path delay in the atmosphere, inverse barometer effects, aliased barotropic response to wind and pressure forcing… Most of these corrections are derived

from weather models such as the European Centre for Medium-Range Weather Forecasts (ECMWF).

However, KaRIn SSHA images sometimes highlight the limit of the atmospheric corrections. To illustrate, Figure 21 shows an example of atmospheric Lee Wave in the Aleutian Islands off Alaska. These internal gravity waves of the atmosphere are caused by the vertical air displacement after the wind flows over a mountain (here the island chains in the North West part of the scene). On the downwind side of the terrain (the center of Figure 21) they cause a periodic change of the pressure, wind

and atmospheric temperature fields. This is the wave-like patterns in panels (b) and (d), that did not exist the day before in panel (a) and (c). Each oscillation is of the order of 10 km, i.e. barely captured by a nadir altimeter, but well observed by KaRIn. Such patterns are frequently observed in the sigma0 surface roughness fields of SAR images, and KaRIn captures the same changes in surface roughness with the periodic changes of the surface wind. KaRIn also shows the impact of these waves in the ocean surface topography.

These atmospheric features are below the spatial resolution of the Level-2 dynamic atmospheric correction (DAC) that are forced by ECMWF winds and pressure. These Lee Wave features exist in many regions, although they might appear randomly depending on surface wind conditions. This is an example of how the joint analysis of the SWOT collocated SAR sigma-0 images and SSHA can help understand what is a small-scale ocean signal, and what may be introduced by corrections that are not yet at the required resolution. These waves will be more frequent in the coastal zones or near islands, so user beware! Once

again, the Level-3 product could serve as a sandbox demonstration for experimental higher-resolution atmospheric corrections (e.g. region DAC correction based on a higher resolution model).

Sometimes, KaRIn images can capture the rapid ocean response to extreme atmospheric signatures in the measured SSHA. Figure 22a show a series of 6 KaRIn daily images from the Calibration Phase, when SWOT sampled the same track daily. Here SWOT crosses the trajectory of the Betty/Mawar cyclone in the tropical West Pacific: 3 days before the cyclone, during

the cyclone itself and 2 days after the cyclone. During the cyclone overflight, KaRIn measurements are strongly affected by

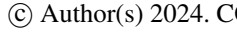



the presence of heavy rain in the arms of the cyclone spiral [7]. At 2-km, the entire cyclone segment becomes very difficult to use. Conversely, the 250-m counterpart in Figure 22b provides more ways to detect the rain regions (blue/red noise) and the regions without rain between the spiral arms (green/blue regions). In the eye of the cyclone, the sky is clear and the topography is pulled higher than the rest of the scene by tens of centimeters. That is likely an inverse barometer response to the cyclone depression, i.e. it should be corrected for by the DAC correction. But the correction does not have the space and time resolution to resolve this extreme event.

The temporal evolution of the background ocean between the left and right panels of Figure 22a (i.e. before/after the cyclone has crossed the KaRIn swath), shows a large warm mesoscale eddy developing before the cyclone, then the SSHA goes down by 20 cm or more (in only 2 days). This is a good example of how a tropical cyclone may siphon up the latent heat in the upper layers of the ocean. This example derived from our Level-3 product gives a very good 2D view of the event, including waves radiating from the cyclone when it is not yet / no longer within the SWOT swath.

Similarly, Figure 22c presents a second example of SWOT crosses an intense wind front (here 20 m/s) in the North Pacific. In this 250-m Level-3 SSHA image, there is a sharp offset of 10 cm over less than 1 km along the storm front. Looking at weather model is not particularly helpful as the model hindcast predicts that the front is in a different position (it travels very quickly). In this example, it is not clear if the KaRIn SSHA offset is real, or caused by a SSH measurement artifact (e.g. lack of resolution of atmospheric path delay, or sea-state bias), or by a deficiency in the DAC correction. The nadir altimeter observes the offset, but its noisier and spread out due to the larger altimeter footprint.

This large frontal offset of 10 cm associated with the wind front could affect some Level-3 algorithms, and in particular the Level-3 empirical calibration: the calibration could interpret the offset as a KaRIn systematic error (bias term B or cross-track slope term L) and in turn create artifacts on the calibration over thousands of kilometers along this specific pass. Due to the magnitude of 10-cm, even a small leakage might be large enough to affect validation metrics.

Exploring these topics is not simple: it requires many different skillsets (remote sensing, atmosphere model, sea-state model, data-driven calibration, etc.). Still, from these examples, it is clear that updating atmospheric corrections with research-grade models might interact with Level-3 algorithms.

## 5.4 Coastal regions

Coastal regions are known to be challenging for nadir altimetry (Vignudelli et al, 2019). The radius of their footprint is often a limiting factor: land can contaminate the altimeter waveform and create various artifacts, which may randomly affect all the retracked parameters. In contrast, SAR Interferometry can extend up to land, and Figure 23 illustrates how 250-m KaRIn measurements provide a seamless SSHA transition from the open ocean to the continental string of islands and then to coastal

---

[7] In this example, we are not using the editing layer, as it would remove most of the scene.





regions. These maps cover the Black Point district in the Bahamas, with very sharp transitions between deep waters (2000 m or more, dark blue in panel a) to very shallow waters (20 to 30 meters, cyan to white in panel a) up to the long island chain in the middle of the scene.

This region has complex circulation patterns, as well as complex wave generation/propagation due to the bathymetry. Contrary to the sparse coverage of nadir altimeters (e.g. a Jason-class mission would provide 2 x 1D lines every 10 days in this scene),
KaRIn captures this complexity with an unprecedented wealth of coverage and precision. In this specific image, KaRIn's 250-m noise is barely noticeable due to low wave conditions; and KaRIn observes the deep water circulation patterns correlated with the bathymetry, with similar patterns in sea surface temperature and ocean color maps from panels (c) and (d). The 250-m SSHA also exhibits very thin lines that are aligned with the shallow underwater dunes of this region.

Importantly, the SSHA is defined up to the last coastal pixel, showing a clear discrepancy between the Western and Eastern
sides of the island chain that are not observed by nadir altimeters (because of their measurement errors very close to the coast). It is very likely that the last few kilometers might be affected by the limited resolution of various geophysical models: the DAC model, the tide models and the MSS models are unlikely to resolve these kilometer-scale or 250-m scale features, since they are derived from lower-resolution nadir altimeter data from previous missions of the past decades.

These limitations cannot be resolved from one image, however an animated sequence of 90 KaRIn images during the 1-day
phase of SWOT highlight that some patterns are time-invariant (i.e. likely geoid error) while the island chain's regional bias is modulated at the tidal aliasing periods (i.e. likely tidal error). The shallow waters are also biased in specific pressure and wind conditions (i.e. maybe DAC errors). New measurements from SWOT are limited by the resolution of these current models, but KaRIn is also (and by far) the best asset to improve the next generation of some geophysical models (e.g. tides, MSS, MDT).

In that context, our Level-3 "unsmoothed" 250-m variant tries to provide all the necessary elements for such expert investigations (e.g. each individual geophysical model that are not included in the 250-m Level-2 product), but does not include all the technical variables related to the instrument: it is a trade-off between convenience for thematic studies and data volume. However, as discussed in the previous sections, it is likely that our editing or calibration procedures might be suboptimal in coastal transitions. More in-depth analyses by coastal and regional experts would be needed to quantify the limitations, and a
collaborative approach would likely make it possible to define good testbeds for our algorithms or to propose improvements in geophysical models.

## 5.5 Hydrology

In their prelaunch assessments of the data-driven calibration, Dibarboure et al (2017, 2022) predicted that the ground segment Level-2 algorithm based on ocean crossovers could be insufficient to meet the requirements during the 1-day phase of the
SWOT mission, because there are very few crossover diamonds with SWOT alone. In that context, our Level-3 multi-mission





calibration was initially designed as a backup algorithm able to meet the Hydrology requirements. Conversely, for the 21-day phase, the crossover L2 algorithm and our multi-mission L3 algorithm were shown to have a similar performance for Hydrology, as the inland interpolation is the primary source of error.

Ubelmann et al (2024) report that both predictions turned out to be true for the Level-2 version "B" (beta pre-validated). So
our research-grade algorithm from section 3.3 was adopted by the SWOT Project as the default Level-2 algorithm for the 1-day phase of SWOT in their reprocessing version "C" (winter 2024). In practice, we delivered the calibration parameters as a by-product of our Level-3 processor, and this correction was ingested by the SWOT Project's Level-2 processor. Conversely, the mono-mission L2 crossover calibration remains the default correction for the 21-day phase, as a SWOT-standalone algorithm has simpler interfaces (no external satellite) than our research-grade processor, and for a very similar output
performance over hydrology surfaces.

In that context, our current Level-3 calibration is by no means optimal for Hydrology, and various improvements could be added: better interpolators for inland or polar segments, better reduction of coastal variability (e.g. tides & DAC leakage on inland water heights), adding inland reference points from stable targets (charted lakes and reservoirs, bridges and roads, etc.). It is also likely that the optimal calibration algorithm for ocean/coastal/estuary users might be different from the optimal
solution for inland rivers and lakes that are not in the tidal region. Exploring these differences in the future will require a wide set of skills from data-driven calibration, coastal ocean physics, hydrology physics and in-situ.

### 5.6   Polar Regions and sea-ice

KaRin's 250-m resolution is generally not needed to study open ocean mesoscale structures, but it becomes a powerful asset in Polar Regions covered by sea-ice. Indeed, the higher resolution makes it possible to retrieve the ocean surface topography
in smaller polynyas, to retrieve surface ice/snow topography, as well as the difference between the ice/snow and ocean surface (sea-ice thickness in the freeboard region).

Figure 24 is a composite between our 2-km "basic" Level-3 product in the Southern Ocean, and our 250-m "unsmoothed" product in the Polar Regions. The zoom in panel (b) shows that our Level-3 calibration is generally decent (consistent colour scheme for the ice-topography), but not perfect in these regions: some passes have a bias of a few centimeters.

Moreover, the sample segment in panel (c) shows the SSHA in m on the left, and the Level-2 sigma0 on the right (arbitrary scale). In this image, the polynyas are easy to recognize from the image texture: water between the ice is visible as blue pixels in the SSHA and white pixels in the sigma0. Note that when there is no wind, or when the wind blows in a direction orthogonal to thin ice cracks, the water surface can be very dark, rather than very bright. In panel (d), we use a simple and arbitrary sigma0 threshold to separate bright and dark pixels. The resulting topography PDF clearly shows two very different populations: the
ice and snow PDF is bell-shaped as expected from the terrain roughness, while the ocean PDF is very peaky and skewed. The



difference between the median values of both distributions would suggest a difference of 15 cm between the surface of the ice/snow and the ocean.

This very simple example captures the potential of KaRIn's 250-m resolution, but it also illustrates how Level-3 algorithm are
important for this type of analysis. Indeed, if the KaRIn topography is skewed by 5 to 10 cm (because of tides model, or data-driven calibration), it becomes extremely difficult to use the image like in Figure 24d. Furthermore, Dibarboure et al (2022) have shown that the Level-2 calibration can be suboptimal in Polar Regions, especially during the wintertime, while the Level-3 is substantially more robust. Similarly, the MSS and tides models have various sources of errors in these regions. To that extent, any improvement with research-grade geophysical models and calibration could make a difference for Polar Regions. Validating such improvements would only be possible with the help of experts from the polar community.

**5.7**   **Future developments and collaborative Level-3 framework**

SWOT and KaRIn demonstrate a breakthrough in the observation of surface water topography. Moreover, our Level-3 processor proves one can integrate the new observing capabilities from SWOT into established altimetry multi-mission processors that have been operating for 3 decades. However, the current KaRIn algorithms and format are still not mature, and will be progressively improved over time. This paper reports a series of Level-2 and Level-3 limitations (e.g. editing in coastal
region, suboptimal calibration in Polar Regions, etc.) which must be addressed in collaboration with various experts (from SWOT and other communities). It is likely that more limitations will be reported by external users and independent validation.

From the very different examples presented above, it is clear that improving SWOT products is not just the sum of independent corrections and algorithms. There are multiple and intertwined problems: sensor processing (nadir & KaRIn), Level-2 geophysical corrections, Level-3 processing layers, regional or surface-specific dynamics that might leak into any item based
on SSHA (e.g. empirical sea-state bias, tide model with assimilation, etc.).

To that extent, it is clear that Level-3 algorithms cannot be developed and improved in a vacuum, with a single homogeneous user community in mind. It is necessary to address thematic needs in tight collaboration with each user community, as well as in collaboration with ground-segment algorithm experts.

In that context, our future developments will focus on three aspects:

1.   We plan to develop and to contribute to open and fair evaluation frameworks for experimental algorithms and corrections (e.g. L2/L3 Ocean Data Challenges[8]).

    2.   We plan to integrate new state-of-art candidates (from all sources, for all algorithm layers) as sandbox or demonstration reprocessing. Our objective is to ease the SWOT product evaluation for users with pre-made and user-

---

[8] https://github.com/ocean-data-challenges





convenient products, as well as to investigate how a given algorithm improvement might be intertwined with other
algorithm layers or specific needs from a given user community.

3.    We plan to explore the possibility and relevance to replace our "one-fit-all" L3 product with community-specific
layers and products (e.g. hydrology, coastal or sea-ice products). It is useful to have continuity from one surface to
another, but the algorithms may need to be adapted and tuned differently.

Our current Level-3 product integrates state-of-the-art research from various research groups. We plan to keep integrating as
many external contributions from the research community as possible, while keeping a flexible but consistent and
comprehensive set of reprocessed products, and near-real time operations.

## 6    Summary and Conclusions

Despite SWOT's unprecedented 2D coverage and precision, the Level-2 products suffer from the same limitations as similar
products from other altimetry missions. They were designed in a standalone framework to meet the mission's primary science
objective. In contrast, some research topics and applications require consistent multi-mission products, such as the Level-3
products provided by the Data Unification and Altimeter Combination System (DUACS) for almost 3 decades, and with 20
different satellites.

In this paper, we describe how we extended the Level-3 algorithms to handle SWOT's unique swath-altimeter data: upgrades
with state-of-the-art Level-2 corrections and models from the research community, a data-driven and statistical approach to
the removal of spurious and suspicious pixels, a multi-satellite calibration process that leverages the strengths of the pre-
existing nadir altimeter constellation, merging of the 1D nadir altimeter and 2D interferometer from SWOT into a single
product... Our qualitative comparisons with other sensors and quantitative comparisons with external nadir altimeters illustrate
the strengths of SWOT/KaRIn and the consistency with other satellites.

We also illustrate and discuss the benefits and relevance of Level-3 swath-altimeter products for various research domains:
basin scale features (e.g. as large as El Niño), global circulation and ocean mesoscale (from ten to hundreds of kilometers),
tides and atmospheric studies, coastal and polar regions. Our Level-3 'basic' and 'unsmoothed' products are more compact
and arguably simpler to use than their Level-2 counterpart, while the Level-3 'expert' variant give more flexibility and control
to those who want to customize it.

More importantly, we illustrate how some data-driven algorithms and models from SWOT may create complex intertwined
problems in a ground-breaking instrument such as KaRIn: leveraging the full potential of 2D images requires diverse set of
expertise and skills, but also a consistent multi-mission and multi-thematic strategy. This has been a strong asset of the Data
Unification and Altimeter Combination System (DUACS). Our current Level-3 swath processor was developed and is now



extending this capability to SWOT, to find a good trade-off between continuously evolving state-of-the-art models, corrections and processing techniques, and consistent reprocessing of past data and forward near-real time products.

**7    Competing interests**

The contact author has declared that none of the authors has any competing interests.

**8    Acknowledgments**

This work was funded by the French Programme d'Investissement d'Avenir [9] to accelerate the adoption of SWOT by the user community, as well as the development of SWOT downstream applications. The authors would like to thank the SWOT Project
from CNES & JPL/NASA for their active help in accelerating the L3 operations during the SWOT Adopt-a-Crossover[10] validation campaigns, as well as many members of the SWOT Science Team for their insights, evaluations and feedback on the first alpha and beta L3 releases.

**9    Appendix**

At the time of this writing, the Level-3 product is available in two releases: version 0.3 was made public on AVISO in
December 2023 (see section 10) and version 1.0 was released is May 2024. The objective of this appendix is to give more details about the content and differences between them.

Firstly, the input Level-2 products are different. This point is illustrated by Figure 25:

- The L3 version 0.3 is based on the Level-2 version "B", also known as PIB0 or "beta pre-validated". The term 'prevalidated' refers to SWOT project timeline: the validation phase of the satellite spans from spring 2023 to spring
2024, and the Project had planned an interim or 'prevalidated' product for public evaluation before the end of the validation phase. The 'beta' term refers to the fact that SWOT products were deemed good enough and distributed a few months ahead of schedule, albeit with known and documented limitations from the ground segment. Our Level-3 product v0.3 inherits from these 'beta' limitations. It also has its own imperfections from the first generation of Level-3 algorithms.

---

[9] https://www.gouvernement.fr/actualite/le-programme-d-investissements-d-avenir
[10] https://www.swot-adac.org/





• The L3 version 1.0 is based on the Level-2 version "C", also known as PIC0 for near real time and PGC0 for reprocessed data. This is the official "pre-validated" (not beta) product [11]. The Level-3 v1.0 product naturally inherits from these ground segment upgrades. Similarly, the reprocessed periods in Figure 25 benefit from a few parameters that were improved/reprocessed between PIC0 and PGC0 (see Table 1).

Lastly, various upgrades of the Level-3 algorithms were also added from v0.3 to v1.0 (this paper describes the 1.0 algorithms):

• We have upgraded the editing layers, added new algorithms for finer outlier detections (more layers and intermediate flag values), and we have added granularity and flexibility to the editing flags. In essence, this change now makes it possible for the user to control a more comprehensive L3 editing process from the 'expert' variant (otherwise, the 'basic L3' is still a good default starting point).

      • We have upgraded the so-called 'phase screen' correction to better mitigate small-scale semi-static calibration
residuals of the order of a few millimeters. This should be transparent for most oceanographers but a good addition for geodesists and all usages where mm-scale systematic biases are relevant.

      • We have improved the L3 calibration mechanism to reconcile the L2 and L3 calibrations: one bias from karin to the swot nadir altimeter, and one bias from swot to the rest of the nadir constellation and Sentinel-6. We have also added a new quality flag for the L3 calibration to isolate outliers and suspicious segments for this algorithm.

• We have retrained the noise reduction UNet for better and more stable performances (re-training, new ocean models with tides, style transfer for measurement errors). Version 1.0 should have fewer artifacts than v0.3.

      • We have tweaked the L3 format to make it more compatible with various standards and generic libraries and tools.

| | Level-3 product v0.3 | Level-3 product v1.0 |
|---|---|---|
| **Input Level-2** | PIA1 before 2023/09/06<br>PIB0 between 2023/09/06 and 2023/11/23<br>PIC0 after 2023/11/23 | PGC0 before 2023/11/23<br>PIC0 after 2023/11/23 |
| **Orbit** | MOE-F | POE-F until 2023/04/30<br>MOE-F after 2023/04/30 |
| **Ionosphere** | NRT GIM model computed from vertical Total Electron Content maps (Chou et al. 2023) rescaled on the orbit altitude with IRI95 model | GIM model computed from vertical Total Electron Content maps (Chou et al. 2023) rescaled on the orbit altitude with IRI95 model (https://irimodel.org/ ) |
| **Wet troposphere** | Model computed from ECMWF Gaussian grids | Model computed from ECMWF Gaussian grids |
| **Sea State Bias** | Non-parametric SSB from AltiKa GDR-F (Tran 2015) | Non-parametric SSB from AltiKa GDR-F (Tran 2015) |
| **Mean Profile/ Mean Sea Surface** | **Hybrid MSS (SIO22,CNES/CLS22,DTU21) (Schaeffer et al. 2023; Laloue et al., 2024)** | **Hybrid MSS (SIO22,CNES/CLS22,DTU21) (Schaeffer et al. 2023; Laloue et al., 2024)** |

---

[11] Since the release of the Level-2 product version "C", the SWOT project has fixed some of the known prevalidated issues. The ground processor upgrade will culminate in the release of a Level-2 version "D", as well as a full mission reprocessing. While these matters will become obsolete in the future, it is currently relevant for L3 users of version 0.3 and 1.0 to check out the public L2 documentation for known limitations and weaknesses.



| | Level-3 product v0.3 | Level-3 product v1.0 |
|---|---|---|
| **Mean Dynamic Topography** | **MDT CNES_CLS_2022** **(Jousset et Mulet 2020; Jousset et al. 2022)** | **MDT CNES_CLS_2022** **(Jousset et Mulet 2020; Jousset et al. 2022)** |
| **Dry troposphere** | Model computed from ECMWF Gaussian grids (new S1 and S2 atmospheric tides are applied) | Model computed from ECMWF Gaussian grids (new S1 and S2 atmospheric tides are applied) |
| **DAC** | DAC v4.0: TUGO forced with ECMWF pressure and wing fields (S1 and S2 were excluded) + inverse barometer computed from rectangular grids (Carrère et al, OSTST 2019) | DAC v4.0: TUGO forced with ECMWF pressure and wing fields (S1 and S2 were excluded) + inverse barometer computed from rectangular grids (Carrère et al, OSTST 2019) |
| **Ocean tide** | **FES2022 (Lyard et al. 2023; Carrère et al. 2023)** | **FES2022 (Lyard et al. 2023; Carrère et al. 2023)** |
| **Internal tide** | HRETv8.1 tidal frequencies: M2, K1, S2, O1 (Zaron 2019) | HRETv8.1 tidal frequencies: M2, K1, S2, O1 (Zaron 2019) |
| **Pole tide** | (Desai, Wahr, et Beckley 2015) & Mean Pole Location | (Desai, Wahr, et Beckley 2015) & Mean Pole Location |
| **Solid earth tide** | Elastic response to tidal potential (Cartwright et Edden 1973; Cartwright et Tayler 1971) | Elastic response to tidal potential (Cartwright et Edden 1973; Cartwright et Tayler 1971) |
| **Loading tide** | FES2022: (Lyard et al. 2024; Loren Carrère et al. 2023) | FES2022: (Lyard et al. 2024; Carrère et al. 2023) |

**Table 1: Input data & altimeter standards used in the L3 KaRIn processor. Items in bold are specific to the L3 product. Other items are inherited from the L2 product.**

## 10   Data availability

All datasets discussed in this manuscript are preserved and distributed by the Agencies' data repositories in compliance with

FAIR requirements (e.g. Core Trust Seal certified, or ongoing certification from the Research Data Alliance).

- The Level-3 SWOT products generated in the frame of this study are distributed by the AVISO repository from CNES from the following DOI:

  - L3 Basic Variant: https://doi.org/10.24400/527896/a01-2023.017
- L3 Expert Variant: https://doi.org/10.24400/527896/a01-2023.018
  - L3 Unsmoothed Variant: https://doi.org/10.24400/527896/A01-2024.003

- The SWOT products used in the manuscript are distributed by mirror centers from NASA and CNES. The SWOT products can be downloaded from either repository. Each SWOT product has a specific product description
document (PDD) and digital object identifier (DOI).

  - NASA: https://podaac.jpl.nasa.gov/swot?tab=datasets is the PODAAC repository for both ocean and hydrology products
  - CNES: https://doi.org/10.24400/527896/a01-2023.016 is the AVISO repository for ocean products
- CNES: https://hydroweb.next.theia-land.fr is the HYDROWEB repository for hydrology products.

- Among the variety of SWOT Level-2 products available on the repositories, this paper specifically uses 2 products :





- o The SWOT Level 2 KaRIn Low Rate Sea Surface Height Data Product, Version 1.1. available on the two centers with two DOIs: https://doi.org/10.24400/527896/a01-2023.015 (AVISO, beta prevalidated collection) or https://doi.org/10.5067/SWOT-SSH-1.1 (PODAAC, collection ID : SWOT_L2_LR_SSH_1.1)
- o The SWOT Level-2 Nadir Altimeter Data Product (AVISO, collection : L2_NALT_IGDR) available from the following DOI https://doi.org/10.24400/527896/a01-2023.005)

- This paper also uses Level-3 products from other altimetry missions (namely Sentinel-3, Jason-3, SARAL), as well as Level-4 gridded products other nadir altimetry satellites, sea surface temperature and sea surface chlorophyll concentration. These datasets are distributed by the Copernicus Marine Service repository : https://marine.copernicus.eu/access-data

- o The Level-3 along-track product is the SEALEVEL_GLO_PHY_L3_MY_008_062 product ID and available from the following DOI : https://doi.org/10.48670/moi-00146
- o The Level-4 gridded product is the SEALEVEL_GLO_PHY_L4_MY_008_047 product ID and available from the following DOI : https://doi.org/10.48670/moi-00148
- o The Chlorophyll concentration is the OCEANCOLOUR_GLO_BGC_L3_NRT_009_101 product ID and available from the following DOI : https://doi.org/10.48670/moi-00278

## 11 Author Contributions

Conceptualization: GD, CU, YF, RM. Data Curation: PP, EC, MR, FB, RC, CA. Formal Analysis: GD, PP, MR, MP, CA, AT. Funding Acquisition: GD, NP. Investigation: GD, RM, PP, CU, MP. Methodology: GD, RM, YF. Project administration: GD, YF, NP. Resources: FB, AD, RC, AT. Software: FB, RC, AD. Supervision: RM, GD, YF. Validation: MP, AD, PP, CU. Visualization: AD, GD, YF, AT, CU. Writing: GD, RM, YF, MP.



**12 Figures & captions**

**Figure 1: Comparison between 1D and 2D altimetry. Panel (a) is a Level-4 map from 1D nadir altimetry (0.25° product from the Copernicus Marine Service, based on 7 nadir altimeter satellites, no graphical smoothing). Panel (b) is our Level-3 product from SWOT in the same region.**




**Figure 2: Overview of the Level-3 algorithm and datasets**





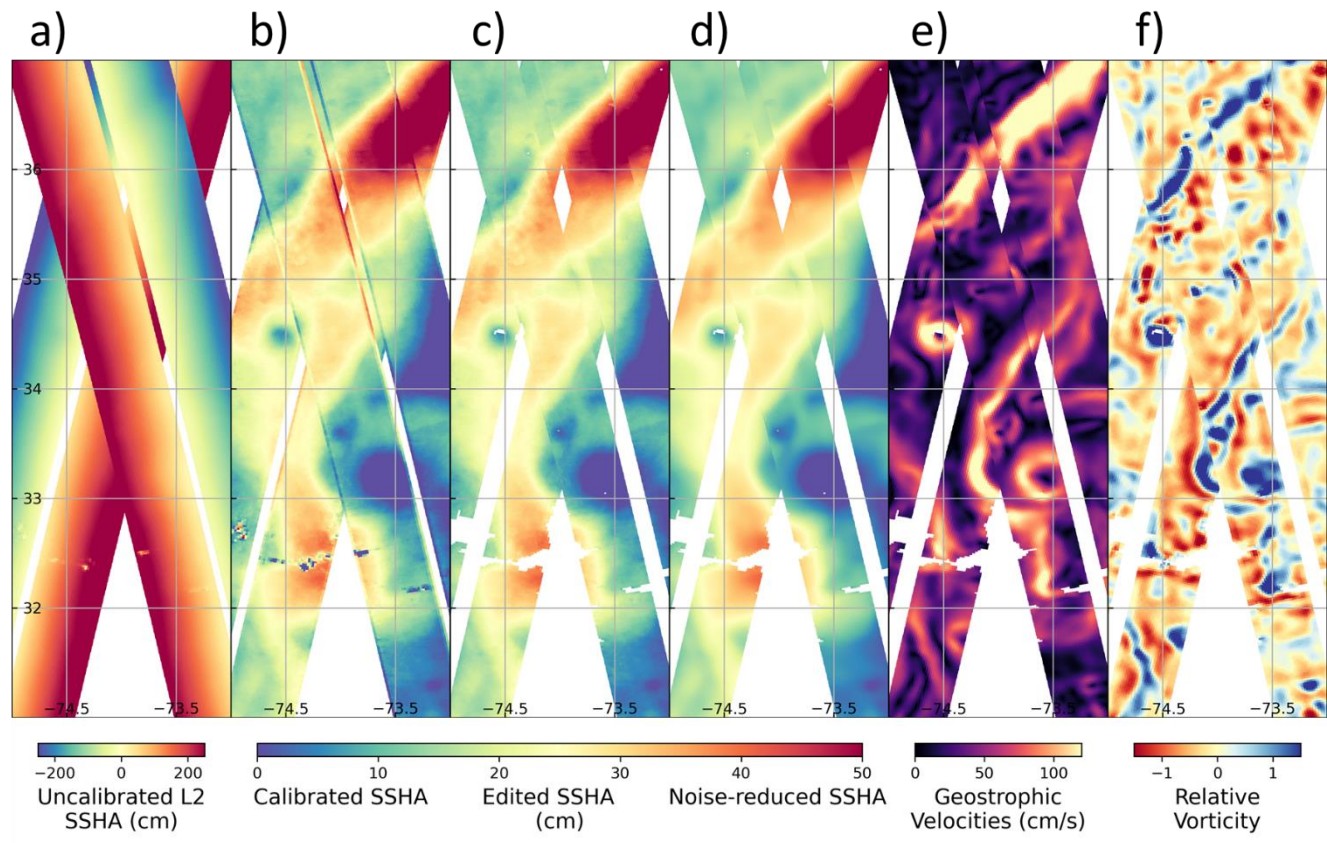


**Figure 3: Example of KaRIn segment (Gulf Stream region) at various steps of the Level-3 processing sequence. Panel (a) is the Level-2 SSHA (uncalibrated) at the beginning of the Level-3 sequence. Panel (b) is the same scene once the Level-3 data-driven calibration is applied. Panel (c) is when the Level-3 editing procedure is applied to get rid of spurious pixels (here heavy rain cells and biased swath edges). Panel (d) is when the noise mitigation algorithm is applied. Panels (e) and (f) are the SSHA derivatives computed from panel (d).**




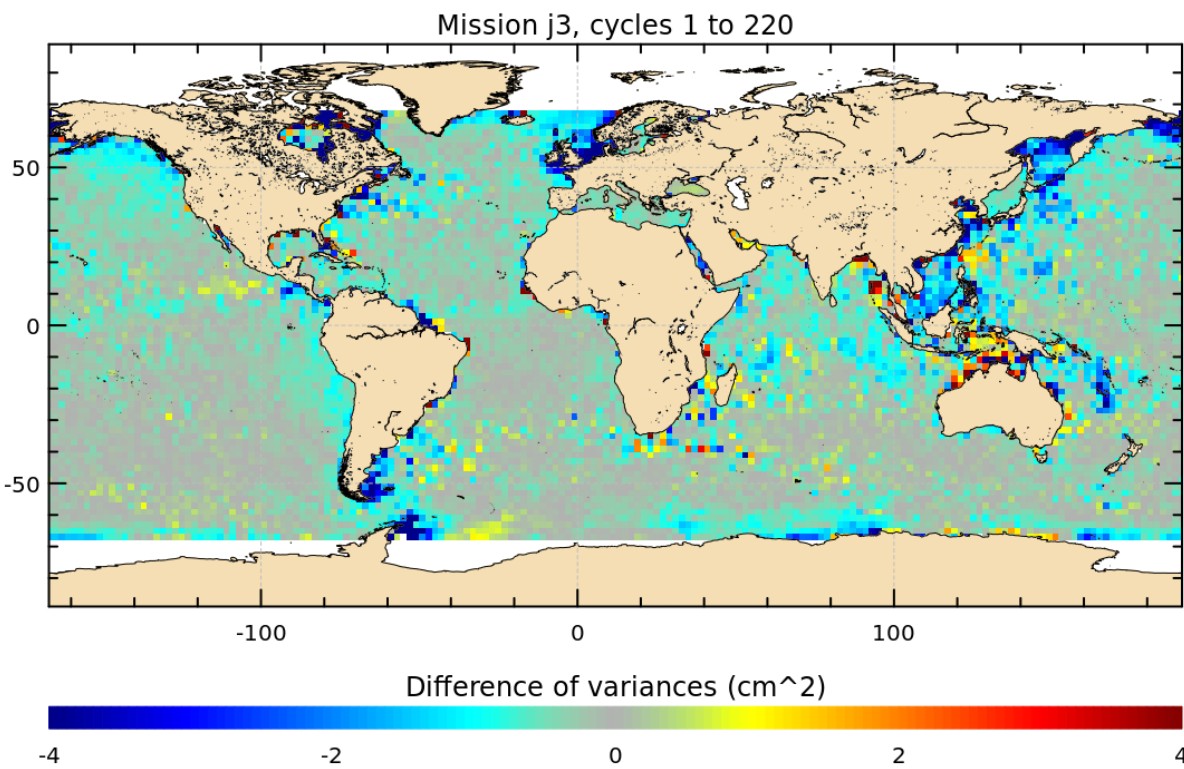

**Figure 4: SSHA Variance difference (FES22 tide correction VS. FES14B tide correction). Negative values indicate that the new FES model provides a better correction of the residual tides signal.**





**Figure 5: Importance and limitations of current Mean Sea Surface Models for SWOT. Panel (a) shows a SWOT/KaRIn SSHA segment in the Caribbean Sea and Tropical Pacific ocean (SWOT 1-day phase) with a zoom north of the Dominican Republic. Panel (b) shows the GEBCO bathymetry in the same region. Panel (c) shows the power spectral density of the KaRIn (thick solid line) and Nadir SSHA (thin solid line) when it is based on the CNES/CLS2015 mean sea surface model (grey); and when the CLS/Scripps/DTU Hybrid 2023 model is used instead (black). The red/blue lines are the estimated errors of each MSS model.**






**Figure 6: Examples of SSHA editing. Left panels are raw SSHA measurements before the editing process. Center panels are the SSHA after the L3 editing process. Right-hand side panels are the editing flag value. Panel (a) is an example of rain cell, located in Gulf Stream (cycle 538, pass 7 and 20). Panel (b) is an example of coastal region, located in the Mediterranean Sea (cycle 538, pass 3 and 16). Panel (c) is an example of internal waves, located off the coast of Brazil (cycle 538, passes 20).**





**Figure 7: Examples of two editing scenes at 2-km (panel a, and c) and their 250-m counterpart (panels b and d). The left panels are the raw SSHA before the editing process. Center panels are the same after the L3 editing process. Right-hand side panels are the editing flag value. Panels (a) and (b) and in the South Pacific Ocean. Panels (c) and (d) are in a sea-ice covered region of the Southern Ocean.**






**Figure 8: Impact of the Level-3 editing layer on the SSHA. Panel (a) shows the percentage of data edited out in the Level-3 product from 10 to 60 km cross-track. Panel (b) shows the variance reduction (unit: cm²) when spurious data are edited out. Note that, in both panels, we ignore eclipse segments.**






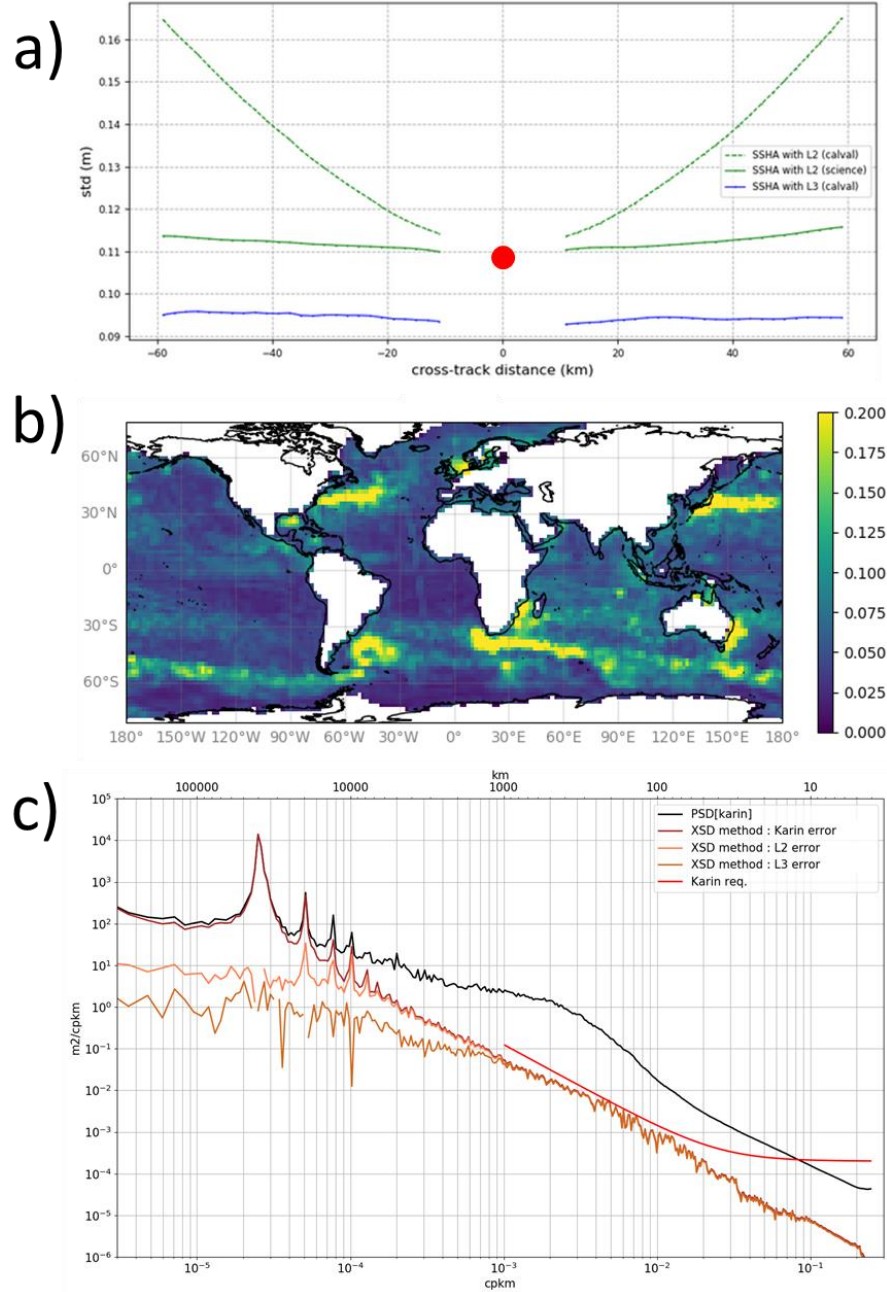

**Figure 9: Reduction of KaRIn's systematic errors with the data-driven calibration. Panel (a) shows the variance of the KaRIn SSHA as a function of the cross-track distance. The dotted green curve is after the Level-2 calibration for the 1-day orbit, and the plain green curve is for the 21-day orbit. The blue curve is after the Level-3 calibration. The red dot is the variance of the nadir altimeter SSHA for the 21-day orbit. Panel (b) is a map of the KaRIn SSHA standard deviation in cm. Panel (c) is the power spectrum of the KaRIn SSHA (black) and an estimate of the KaRIn systematic errors (B/L/Q components) before calibration (dark brown), after Level-2 calibration (orange), and after Level-3 calibration (light brown). The KaRIn ocean requirements are also shown in red.**






**Figure 10: Example of 2-km KaRIn SSHA before (top) and after (bottom) denoising in the Gulf Stream (May 2023). From left to right: sea surface height anomaly, geostrophic velocity, and relative vorticity.**





**Figure 11: Same as Figure 10 for the 250-m unsmoothed L3 product.
The left column is the SSHA and the right column is the geostrophic velocity.**




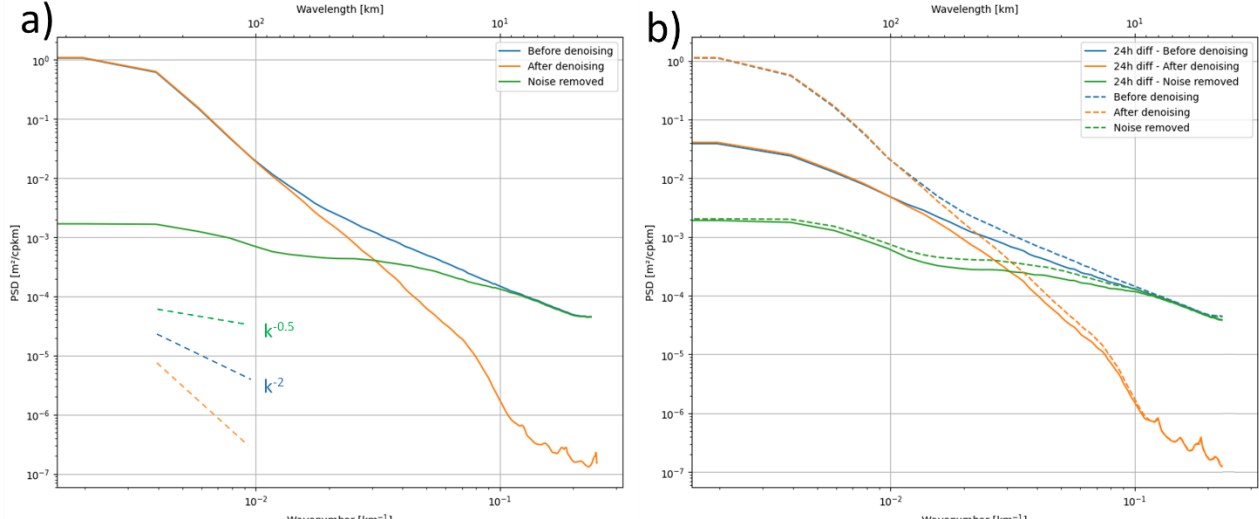

**Figure 12: PSD of the KaRIn SSHA in m²/cpkm. The blue curve is before our L3 denoising. The orange curve is after the U-Net is used. The green curve is the PSD of the "noise" removed by the U-Net. Panel (a) is for the 2-km resolution. Panel (b) is the same as panel (a) but adding 24-hour differences from the 1-day phase (plain lines).**






**Figure 13: Example of Level-3 SSHA composite (unit: cm) for November 2023 (ascending passes).**
**Subpanels (a) to (i) are regional zooms from the global map.**








**Figure 14: Same as Figure 13g in the Gulf Stream region. Panel (a) is the SSHA in cm.
Panel (b) is the corresponding geostrophic velocity in cm/s.**





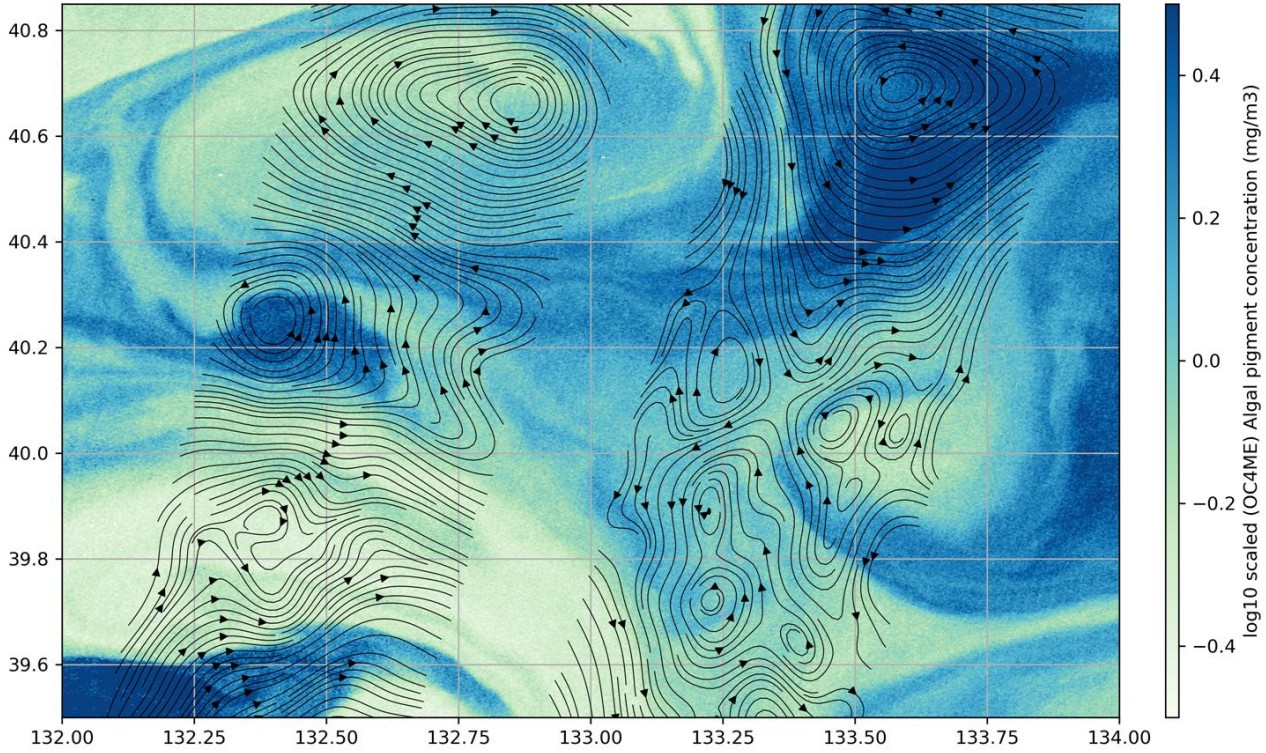

**Figure 15: Comparison between Sentinel-3 / OLCI sensor and SWOT KaRIn for May 11, 2023. The background image is the Chlorophyll OC4ME product from the Copernicus Marine Service. The black lines are streamlines derived from the SWOT/KaRIn geostrophic velocities (2 swaths of 50 km each).**



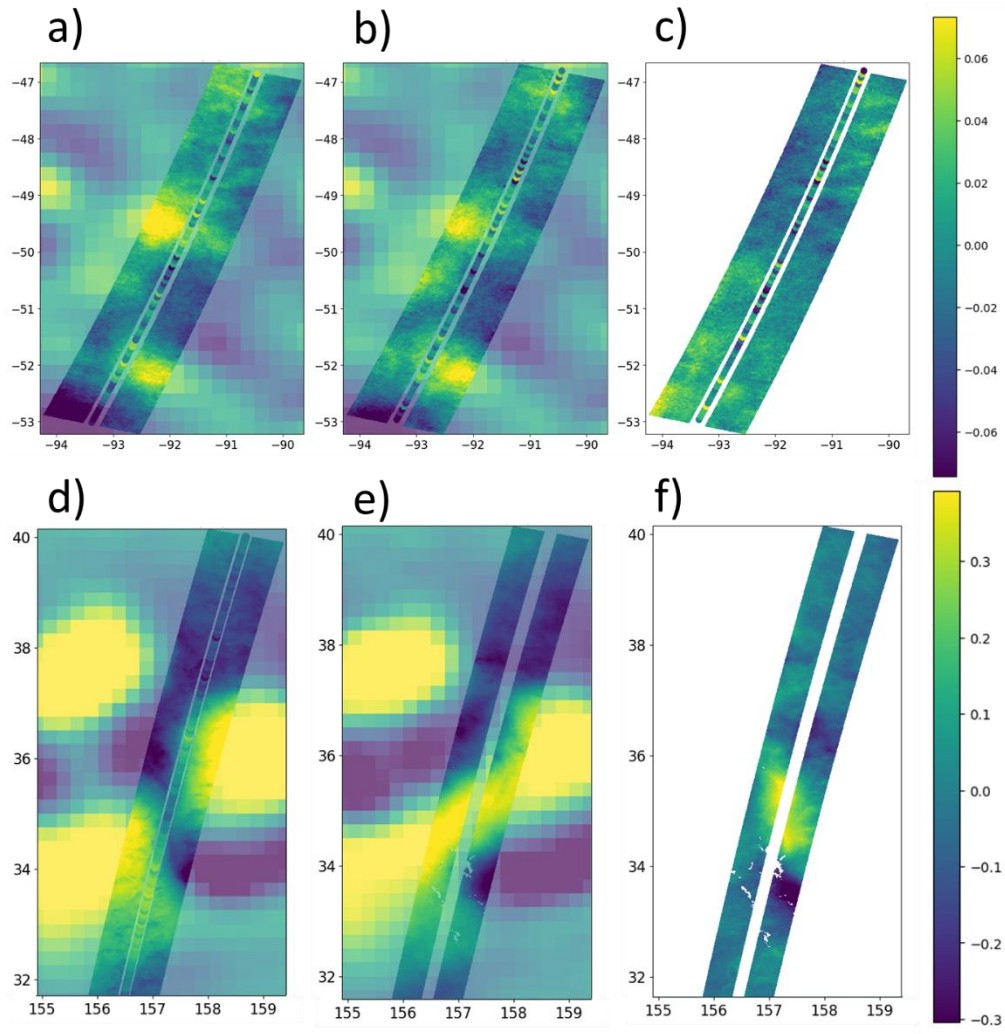

**Figure 16: Consistency of the Level-3 product in time and with nadir altimetry in the Southern Ocean (top panels) and in the Kuroshio region (bottom panels). Panels (a) and (d) shows an arbitrary KaRIn swath during the 1-day phase of SWOT. The KaRIn image is shown on top of the Level-4 maps from the Copernicus Marine Service (pixelated background, same colour scale). Panels (b) and (e) are the same regions, 4 days later. Panels (c) and (f) are the difference between the left and right columns.**




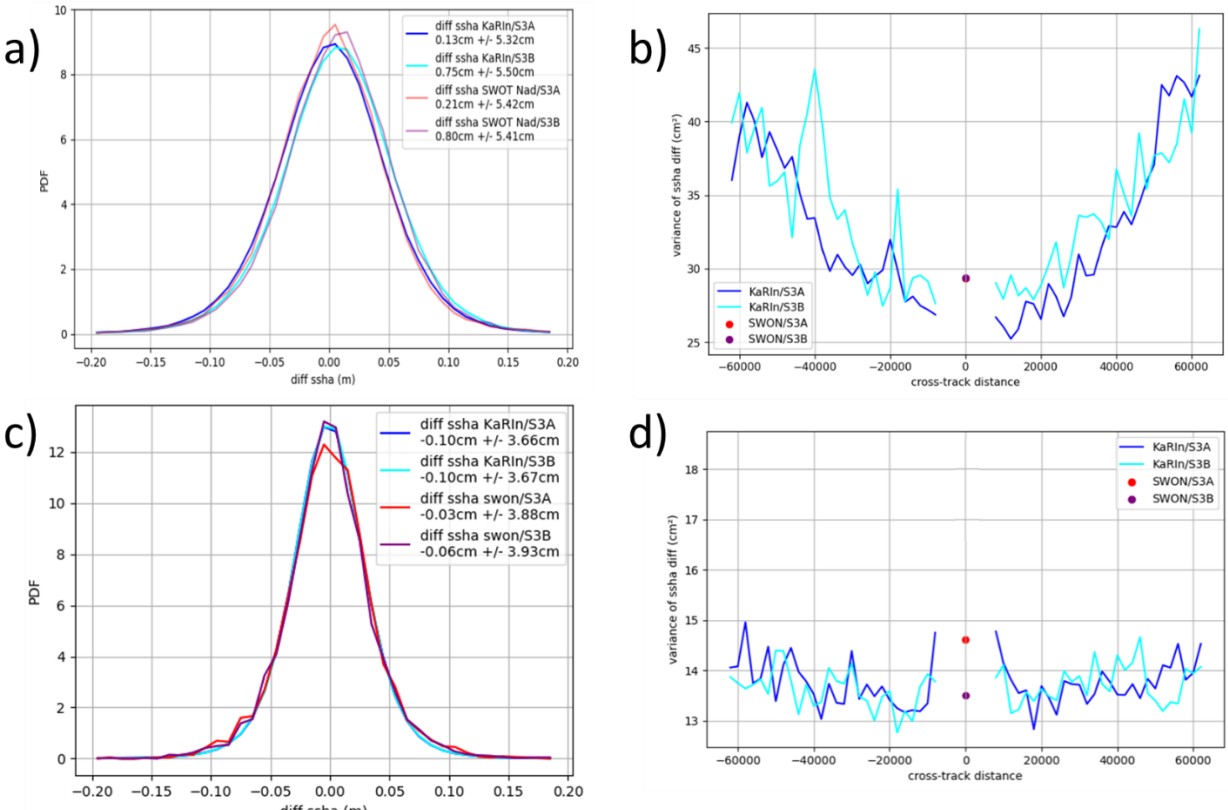


**Figure 17: Statistics of the crossover differences between the KaRIn SSHA and the Sentinel-3 SSHA for the Level-2 products (top panels) and the Level-3 products (bottom panels). Panels (a) and (c) are the probability distribution functions of the SSHA difference. Panels (b) and (d) are the variance of the SSHA difference as a function of the cross-track distance. Dark blue is for SWOT/KaRIn and Sentinel-3A. Light blue is for SWOT/KaRIn and Sentinel-3B. Red is for SWOT/nadir altimeter and Sentinel-3A. Purple is for SWOT/nadir altimeter and Sentinel-3B. The crossover time difference is 24 hours or less.**




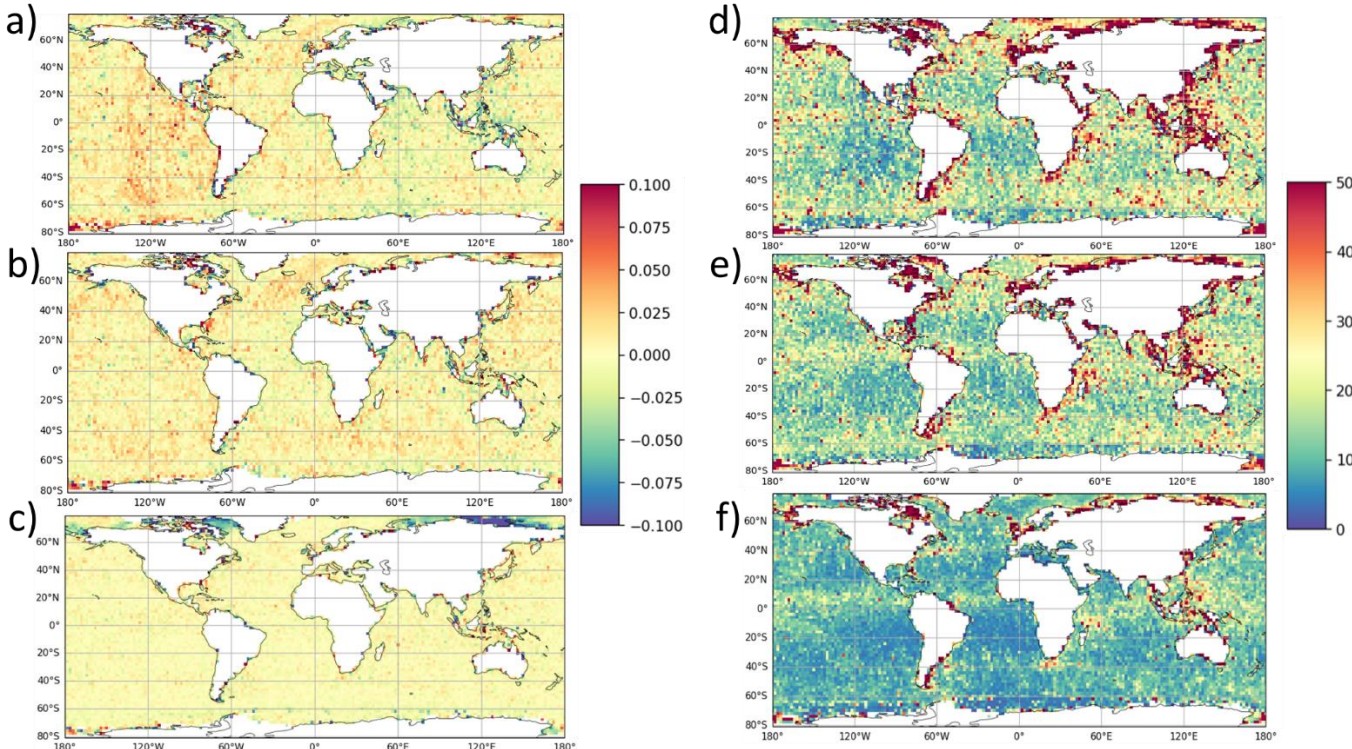

**Figure 18: Crossover SSHA differences between KaRIn and Sentinel-3 A&B, over November 2023 to February 2024, with crossover time differences of 24h or less. Panel (a) is the mean difference for Level-2 products of both missions (unit: m). Panel (b) is the mean difference for Level-2 KaRIn products and Level-3 Sentinel-3 products from the Copernicus Marine Service. Panel (c) is the mean difference between our Level-3 KaRIn products and Level-3 Sentinel-3 products from the Copernicus Marine Service. Panel (d) is the variance of the difference between Level-2 products in cm². Panel (e) is the variance of the difference for Level-2 KaRIn products and Level-3 Sentinel-3 products from the Copernicus Marine Service. Panel (f) is the variance of the difference for Level-3 products.**



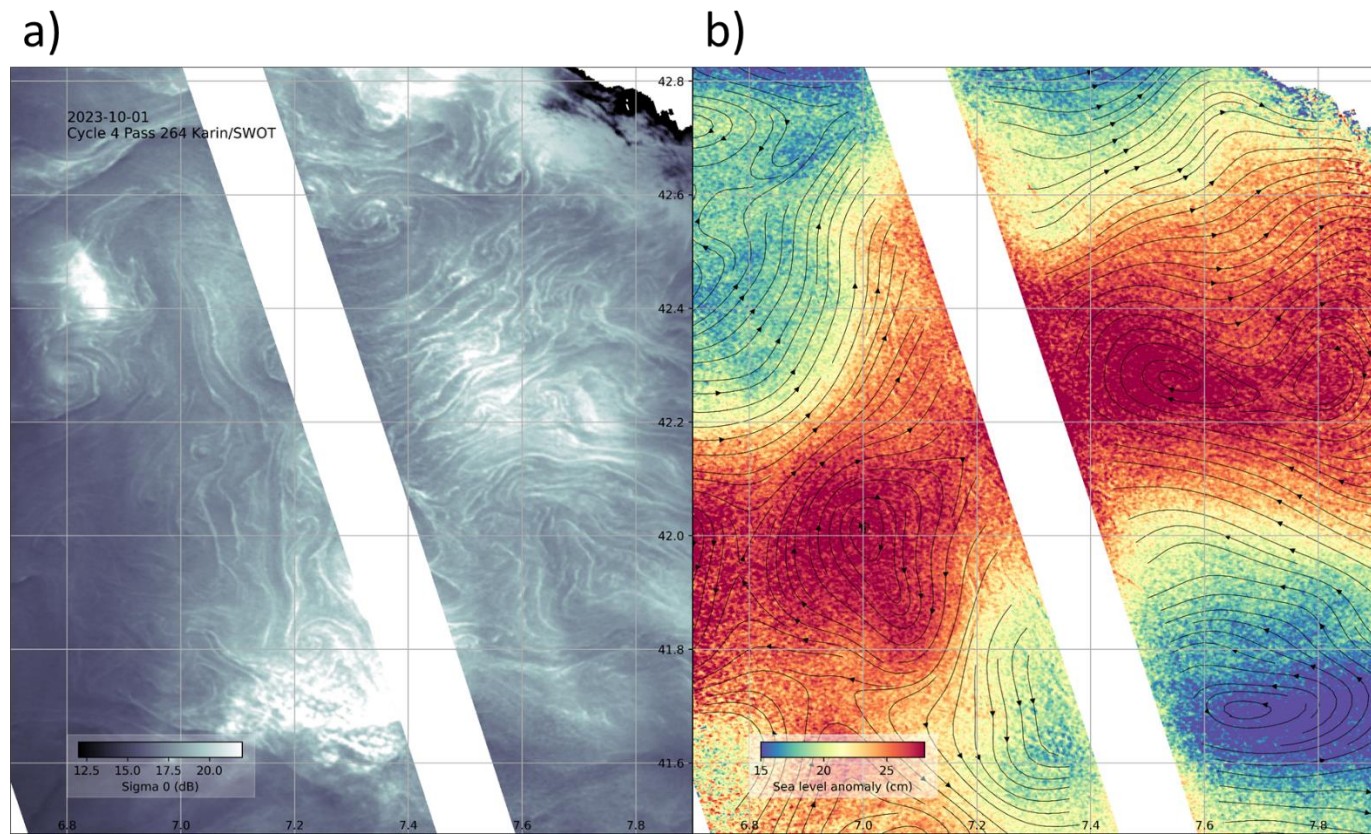

**Figure 19: Small to submesoscale from KaRIn measurements (October 2023, Mediterranean Sea). Panel (a) is the sigma0 in dB from the Level-2 products. Panel (b) is the 250-m SSHA in cm. The black lines in panel (b) are streamlines derived from 2-km geostrophic velocities.**








**Figure 20: Signature of internal tides in the Level-3 product.** Panel (a) is the KaRIn SSH (unit: cm) for the 1-day orbit in the North Atlantic Ocean with a transition between smooth bathymetry (right hand side segment), to rugged bathymetry (left hand size segment). Panel (b) is a weekly coverage for the 21-day orbit over the Mascarene Plateau (isobaths in black). Circular internal tides are radiating from the gap (black arrow) between the Saya de Malha Bank and the Nazareth Bank.





**Figure 21: Example of SWOT Level-3 SSHA over an atmospheric Lee Wave (Aleutian Islands). The top panels are the L2 sigma0 from KaRIn (linear), and the bottom panels are the KaRIn SSHA in m. The left and right panels are 24 hours apart.**





**Figure 22:** KaRIn Level-3 topography when SWOT crosses the trajectory of the Mawar/Betty cyclone during the 1-day phase of SWOT. Panel (a) is a time series of six subsequent days of the same region (3 days before the cyclone, and 2 days after the cyclone). Panel (b) is the cyclone overflight seen at 250-m resolution (unit: cm). Panel (c) is an example of 250-m SSHA (unit: m) when SWOT crosses an intense wind front (here 20 m/s) in the North Pacific Ocean (cycle 2, pass 41).





**Figure 23: Overview of the SWOT measurements at Black Point district in Bahamas. Panel (a) is ESRI optical imagery: dark blue regions have bathymetry of 2000 m or more, and cyan to white regions are shallow waters of the order of tens of meters. Panel (b) is the 250-m KaRIN SSHA in cm. Panels (b) and (c) are Ocean Colour and Sea Surface Temperature maps from the Copernicus Marine Service.**








**Figure 24: KaRIn SSHA in the Southern Polar Region. Panel (a) maps the eddy field in the Antarctic Circumpolar Current and the sea-ice topography closer to Antarctica. Panel (b) is a zoom in the western part of the sea-ice coverage. Panel (c) is an arbitrary segment KaRIn segment over sea-ice. The 250-m KaRIn Level-3 SSHA in cm is on the left and KaRIn Level-2 sigma0 (linear) is on the right. Panel (d) shows the height PDF for bright and dark targets of panel c (arbitrary threshold on the sigma0 per pixel).**




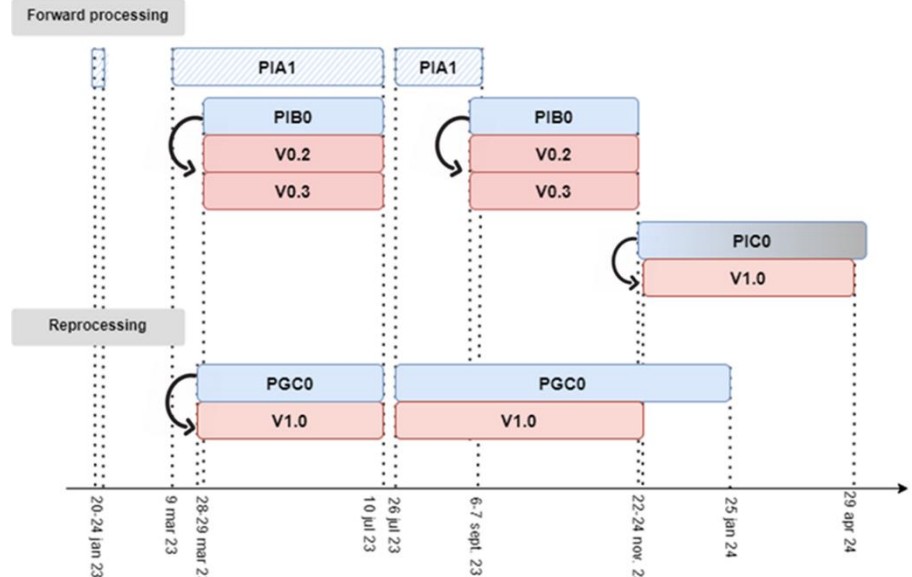

**Figure 25: Schematics of the Level-2 and Level-3 product versions and their temporal coverage at the time of this writing. The blue boxes are the Level-2 products from the SWOT Project, and the red boxes are our corresponding Level-3 product.**

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
