# Peer review of "Blending 2D topography images from SWOT into the altimeter constellation with the Level-3 multi-mission DUACS system"

_EGUsphere, 2024_

## Author Response (AR1)

**RESPONSE TO RC1**

The authors would like to thank Reviewer #1 (RC1) for their positive evaluation on the product and manuscript, and for their in-depth feedback. The revised manuscript accounts for each detailed comment from RC1 as follows (see detailed comments in the full version with track changes).

*[RC1] The abstract does not specify that this paper addresses LR or ocean data.*

[REPLY] Good point, we need to clarify this (e.g. for coastal/cryosphere communities as they might want to compare LR and HR products).

*[RC1] The paper says LR means "low resolution". SWOT project documentation is inconsistent on this point and sometimes says it means "low rate". (No action requested.)*

[REPLY] Probably a minor point but we will add a footnote just in case, and maybe a few words about key differences between LR and HR products, as they are vastly different. The Project has also released a new handbook document which we should definitely include in our reference list.

*[RC1] Line 244: is a comma missing after eclipses?*

[REPLY] No. The revised manuscript will be reworded as "eclipse transitions", with no 's' and no comma. When the satellite transitions from illuminated by the sun to eclipse shadow (or the opposite), there is a substantial change in thermal conditions for the instrument. In the current product release (Level-2 version C), the SWOT Project flags about 2 minutes of data after each transition, as a buffer for the instrument to stabilize. Recent studies show that this is unnecessarily conservative and the eclipse transitions will be retrieved in version 2.0 of our Level-3 products (and maybe Level-2 version D).

*[RC1] Lines 354 and 357: The meaning of "ground segment" may not be obvious to many users. I think this refers to mission ground processing*

[REPLY] That is a good point. The review manuscript will introduce this term with a footnote. Ground segment indeed refers to the processing software operated by the Space Agencies on the ground. This software processes raw satellite telemetry into final science products.

*[RC1] Line 511: the figure does not really show propagation of El Nino, in the sense of a signal moving from one place to another. It just shows a change, which may be associated with El Nino or some other thing that changed during the sampling period.*
*[RC1] Lines 514-522: Thank you! This explanation is helpful!*

[REPLY] The full explanation in lines 514+ is clearer than the topic introduction in line 511. The revised manuscript will be adjusted in line 511 for the sake of clarity.

*[RC1] Figure 15: While figures are well labeled with lat/lon, it would be nice to state the geographic regions in the figure captions. This would just help the reader know the context.*

[REPLY] Agreed. This will be added in the figure caption of the revised manuscript.

*[RC1] Lines 582-583: "4 eddies in the SWOT scene"--> Is this referring to the Kuroshio case? I don't see four eddies in the SWOT scene.*

[REPLY] Yes, this is the Kuroshio case. This will be clarified in the revised manuscript.

*[RC1] Lines 616-619: Something seems wrong, or I am confused about the labels or what is being said. I think these should say S3B/SWOT-nadir instead of S3B crossovers and S3B/KaRIn instead of KaRIn crossovers.*

[REPLY] This paragraph is poorly worded and indeed confusing. The goal is to compare SWOT$_{NADIR}$ or SWOT$_{KARIN}$ with S3A or S3B, but the sentence isn't very clear about which is which. The point is that both instruments / technologies from swot yield almost zero bias (as opposed to a small bias from S3B), and a similar standard deviation. This will be clarified in the revised manuscript.

*[RC1] Lines 634-636: It isn't obvious how this conclusion was reached.*

[REPLY] These details are somewhat beyond the scope of this paper, but we agree this point needs to be clarified. We will add some clarifications in the revised manuscript. For the global ocean and the science orbit, the geophysical SSHA variance has no reason to be different in the nadir, near-range or far-range areas. So the plot should be flat as a function of the cross-track distance. In contrast, the root cause of the so-called systematic errors (e.g. roll or phase errors) induce a cross-track dependence with linear and quadratic amplitude (i.e. $x^2$ or $x^4$ in variance). The parabolic shape of the variance as a function of the cross-track is a clear indicator of residual systematic errors that have not been fully calibrated. It is also a simple way to estimate the average residual errors (which are noticeable in L2 and very small in L3).

*[RC1] Lines 814-815: I tend to doubt that a tropical cyclone could remove enough heat to lower sea level by 20 cm. It seems more likely a dynamic response caused an adjustment in SSH (ie, the heat probably moved laterally).*

[REPLY] Interesting point. This would be large but not impossible as per some published reports. That said, it could very well be caused by other effects, such as lateral movements, or even measurements errors (e.g. sea-state bias residuals in the cyclone wake). We will reword this sentence in the revised manuscript.

We will also fix the following typos (and poor English articulation) as suggested by RC1 :

*[RC1] Line 174: I suggest "tide models" instead of "tides models".*

*[RC1] Lines 229-230: suggest replacing "images, which are perfectly fine" with "good data".*

*[RC1] Line 292: Text refers to the tropical Pacific Ocean, but 41S is not in the tropics. It should say subtropical South Pacific.*

*[RC1] Line 329: suggest adding "that" between "highlights" and "it is essential"*

*[RC1] Line 339: no comma needed after users*

*[RC1] Line 594: ...larger scales ARE not significantly degraded.*

*[RC1] Line 696: suggest "limitations" instead of "limits".*

*[RC1] Line 702: suggest "view from KaRIn" and "large-scale, basin-wide"*

*[RC1] Line 703: no hyphen needed in El Nino.*

*[RC1] Line 719: I think the word "but" should be deleted.*

*[RC1] Line 741: suggest "tidal residuals"*

*[RC1] Line 747L missing letter "a"-- "tides have a very different paradigm"*

*[RC1]* Line 910: missing letter "s" on first "regions"

*[RC1] Line 926: suggest "one-size-fits-all"*

*[RC1]  Line 950: suggest "requires a diverse..."*

**RESPONSE TO RC3**

The authors would like to thank Reviewer #2 for their evaluation of the manuscript (RC3) and associated products. We also appreciate their helpful in-depth feedback to try and improve the quality and accessibility of this paper.

The main criticism is the manuscript length (text and number of panels), which is discussed below.

RC3 also requests a series of relatively minor clarifications and sentence rewording. These will be addressed in the revised manuscript (see below).

Also, it seems Reviewer #2 had a PDF-converted document with some corrupted figures (transparent background became black), whereas we thought the issue had been fixed with the editorial team (the online version seems ok). This will be addressed in the revised manuscript.

The revised manuscript accounts for each detailed comment from RC3 as follows (see detailed comments in the full version with track changes).

*[RC3] While this manuscript will be of interest to the ocean research community, it requires significant author editorial revision before it can be accepted for publication. It is a very long manuscript and requires a significant review to improve the writing and presentation. For example, the value of the inclusion of section 4.1 Qualitative Assessment is not clear. Section 5 provide a much more informative information on the benefits and advances that will be derived from SWOT. Given this, it is suggested that the authors remove section 4.1. Section 4 will then be a Validation section with only (Section 4.2) that provides a quantitative assessment of the performance of the SWOT derived products.*

[REPLY] This main comment is understood yet very difficult to account for. We do admit it is a long and detailed manuscript.

SWOT/KaRIn brings major changes to the Ocean Surface Topography observations. That comes with an ability to resolve new ocean features, but also with some significant limitations, caveats and processing changes. While writing the first draft manuscript, we tested two options. The first one was to keep it short and focused on our L3 processing and validation (e.g. section 3 and 4.2 only). That option came out as a suboptimal: the relevance of our work/product was difficult to understand for readers that were not familiar with SWOT (mostly because some SWOT properties/issues were not introduced). So we opted for the second option: to put our work and product into a broader context, and to take the time to introduce what are the strengths and limits of our products, and how they are related with SWOT in general. We ended up with a long manuscript and many figures, albeit exceedingly shorter than other SWOT documents (e.g. hundreds of pages just for the handbook and product description documents). Our internal reviews from SWOT and non-SWOT readers were much more positive because the manuscript was deemed much clearer and our assertions were put into their context.

This difficulty was also mentioned by RC1 (difficulty to introduce SWOT product, comparison with other SWOT documents). RC1 felt that this manuscript has a good balance (accessible yet precise for SWOT newcomers). Therefore, we believe it is difficult to prune entire sections from the manuscript unless we lose significant and useful information on the product, features and limits.

More specifically, we think section 4.1 is very important for the flow of the manuscript: it introduces many basic features that are specific to KaRIn (e.g. oceanic features not observed by nadir altimetry, temporal sampling pattern such as the El Nino stroboscopic discontinuities) or to our Level-3 processing (e.g. ability to better retrieve the larger ocean scales). In contrast, section 5 is here to warn potential SWOT users about the limits and risks associated with some processing and corrections. Removing either section to get a substantially shorter manuscript would be easy for us, but we think the resulting shorter manuscript would be less precise and less clear, and sometimes misleading.

Merging sections 4.1 and 5 (i.e. algorithm, validation, discussion of the strengths and limits) might be different way forward, but that would not yield a significantly shorter manuscript.

*[RC3] The readability of the manuscript would be greatly improved, if the authors revised the text to remove the many instances where information presented begins with "Figure x/Panel X shows….". For example line 205 is currently written as "Panel (a) shows the SSHA from KaRIn: the zoom from 20 to 22°N exhibits some very unusual features for SSH anomalies." This could be revised to "The SSHA from KaRIn (figure 5a and ,sub-panel from 20 to 22°N) exhibits some very unusual features for SSH anomalies. This rewrite focusses the reader on the SWOT product and how it shows new features of the ocean that is supported by reference to an appropriate figure.*

[REPLY] This point is taken. The revised manuscript will try to reword some or all occurrences listed in RC3. We would like to thank Reviewer #2 for their many detailed suggestions below.

*[RC3] The other major issue is the number of figures and their production quality, and how they are referenced in the text. There are 25 figures, many with numerous sub-plots. Are all these figures essential to the manuscript? For some sections multiple figures are provided to highlight various aspects of decision made in the processing step. While it is appropriate to discuss in the text the impact and reasoning behind processing steps, it is not clear that these require a stand-alone figure. I suggest the authors consider what figures are essential and which figures are "nice to have" but not essential and can be removed, or if features of the processing step or when discussing the relevance and limitation of the dataset (section 5) can be shown by a carefully chosen region such that s figure can be referred to multiple times to highlight impacts of different processing steps and points in section 5. For example, I suggest you consider only showing one figure or combine some panels of figure 5 and 6 into one figure that shows pertinent examples the impact of editing at the various spatial scales and or regions and (Lines 273-306) I'm not sure of the value of showing two examples of result of the editing process steps (Figures 6 and 7) and the associated text that discuss each panel.*

[REPLY] Like for the manuscript length, we tried to pick individual figures to illustrate, in a simple way, each specific assertion (strength or weakness, from KaRIn or from our Level-3 product). To illustrate, each topic from section 5 has their own picture (or re-uses previous ones). The advantage is that each item can be clearly identified by SWOT newcomers in these products.

Specifically for Fig 5 & 6, the first panels for Figure 5 illustrate what geoid artifacts might look like in KaRIn images (so they can be identified by SWOT users), while the last panel gives a quantitative global assessment (to explain why the new MSS model is critical for KaRIn but not nadir altimeters). Similarly, the different lines of Figure 6 illustrate different features (eclipses, rain, internal tides): merging these features into a single panel makes it more difficult to describe and

disentangle these effects for SWOT newcomers. We could prune some items altogether (e.g. no figure/discussion on geoid or tides) and refer to external papers (in prep or rev). Yet, in turn, this would make our manuscript shorter but more difficult to tackle.

That said, this review comment is understood: we will try to prune some panels / figures, and unnecessary sentences in the revised manuscript.

*[RC3] The production quality of the figures is not appropriate. All figures need to be revised to ensure that they meet OS production standards. Figures must be labelled (sub-plots, x-, y- axis) and font size is legible, and black background removed. Also Figure caption details/information should be removed from the main text and only included only in the figure captions.*
*[RC3] Revise all and produce production quality figures.*

[REPLY] It seems the quality some pictures was broken by an automated word to pdf conversion with the journal template (e.g. the transparent background was turned into a black background, thus hiding panel or axis labels). The default PDF compression may have degraded the quality of some pictures. We thought this had been fixed with the editorial team, and the online PDF looks okay: https://egusphere.copernicus.org/preprints/2024/egusphere-2024-1501/egusphere-2024-1501.pdf

We will update the revised manuscript with non-transparent pictures.

*[RC3] line 16 What is meant be standalone ground-segment. Is ground-segment the same as ground track? To the non-expert this may not have a clear meaning. I suggest rewording that non-experts will understand.*
*[RC3] I note RC1 also had a question regarding the meaning of ground-segment. The author have indicated they will add a footnote to explain. The footnote should be referenced when the term is first used*

[REPLY] That is a good point. The review manuscript will introduce this term with a footnote. Ground segment indeed refers to the processing software operated by the Space Agencies on the ground. This software processes raw satellite telemetry into final science products.

*[RC3] line 34. To this sentence add information dates, including when SWOT was launched and time period it operated in the 1-day repeat orbit.*

[REPLY] Good point. This will be added in the revised manuscript.

*[RC3] Lines 510-513 For clear interpretation of this paragraph - consistency of ascending passes and propagation of warn pacific warm pool - the x-axis (time) need to be defined in panels a-i*
*[RC3] Figure 13 - axes need to be defined on the sub-panels. Given these panel are ascending passes is the x-axis a longitude/time varying coordinate?*

[REPLY] Actually, each thumbnail is just a zoom from the global map. The x-axis is longitude, and y-axis is longitude. However, because of the way SWOT's orbit scans the global ocean, the x-axis is indeed related to time, albeit in a complex way (as explained lines 514-522). We think this property is one of the reasons section 4.1 is relevant. That said, we will update the caption of figure 13 to make it less confusing.

*[RC3] Line 537. Is unclear what new information with respect to the qualitative assessment is shown in figure 14 with respect to Figure 13. Are both figures required?*
*[RC3] Figure 14 caption is incorrect this is not the same as figure 13 g.*

[REPLY] Good point. Panel (a) is admittedly redundant with the global map and some thumbnails (the region is different, but the message they convey is the same). We could probably prune this panel. Conversely, Panel (b) is the only one to introduce SSH derivatives and its caveats on SWOT (one cannot simply use SSH gradients due to measurement errors and/or internal wave content). We think panel (b) is a useful asset to keep in the revised manuscript.

*[RC3] Lines 564-568 Much of this information should only be included in the figure caption and should not be provided in the manuscript text.*
*[RC3] lines 611-615 Much of this information should only be provided in the figure caption.*
*[RC3] lines 671-688. These paragraphs and figure 18 need to be revised. It is unclear what region in the eastern Pacific Ocean and Indian Ocean has the large differences. Why is figure 18 for the entire global and not for specific regions that are of interest. There are sentences in the text that should be contained in the figure caption*
*[RC3] Please ensure all caption provide a comprehensive description of the figure and subplots. Remove all caption information from the manuscript text.*

[REPLY] We will try to streamline the text and to move description details to the figure captions. The maps from Fig18 are global to illustrate the difference between L2 and L3 as the variance is not homogeneous. We think the geographical distribution of the variance is needed because it is a good indicator of some error causes (e.g.: rain & L3 editing).

*[RC3] Line 706 What is sigma-0?*

[REPLY] Sigma0 is the radar power backscattered by the surface (also known as NRCS, or radar cross-section in other communities). It essentially provides information on the surface roughness at the radar wavelength scale (i.e. wind and wave) and/or the effect of atmospheric attenuation (e.g. by rain). We will add a footnote in the revised manuscript to introduce this term.

We will also fix the following typos and poor English articulations as suggested by RC3 :

*[RC3] Within the manuscript remove "…" and ",etc" at the end of lists and replace with "for example, [list of examples].*

*[RC3] line 28. change operations to operation.*

*[RC3] Line 47 Change "...., as small as a few kilometers (Figure 1).*

*[RC3] line 47. Delete sentence beginning "Figure 1.....*

*[RC3] line 48. Remove Panel (a) ..... constellation (7 satellites).  This information should be included in the figure caption and not repeated in the manuscript text*

*[RC3] line 52 Change "large eddies that were resolved in panel (a) .." to "large eddies that were resolved by the 1-D altimetry profiles (FIgure 1a), are .."*

*[RC3] line 56 add (Figure 1b) to end of sentence*

*[RC3] line 57. Change spring to months, or change to Austral Spring or Boreal Spring so the reader has some idea of actual months*

*[RC3] line 68 Change ".., as well as a prone.." to ".., as well as being prone ..."*

*[RC3] Line 101. "Figure 2 ...." It is suggested that this sentence is moved to beginning of next paragraph and rewritten as "An overview of the SWOT algorithm sequence is presented in figure 2, which has two ... components."*

*[RC3] Line 102, Change "...is the left-hand side grey block…" to "..(figure 2, left-hand side) is…"*

*[RC3] line 111-114. Can this be added to the end of the previous paragraph.*

*[RC3] Line 129-130. I suggest a revision of the sentence "The noise-reduction...." to focus on the ability to calculation of meaningful derived ocean quantities that show the small-scale ocean features. For example, " The noise-reduction algorithm makes it possible to use the SSHA to calculate derived quantitative such as geostrophic velocity and vorticity that capture the small-scale ocean features (Figure 3 e and f).*

*[RC3] Line 147 change "tides models" to "tidal models" and elsewhere in the manuscript where appropriate (eg. line 174)*

*[RC3] line 205 Change "Panel (a) shows the SSHA from KaRIn: the zoom from 20 to 22°N exhibits some very unusual features for SSH anomalies." to "The SSHA from KaRIn (figure 5a and ,sub-panel from 20 to 22°N) exhibits some very unusual features for SSH anomalies.*

*[RC3] Line 205. Delete "Panel (a) shows the SSHA from KaRIn: the zoom from 20 to 22°N exhibits some very unusual features for SSH anomalies."*

*[RC3] Line 206. change ".... bathymetry: " to "Bathymetry (Figure 5b):"*

*[RC3] line 208 "..... wherever the bathymetry is rugged: uncharted seamount, rifts, continental shelf ..." to " ... wherever the bathymetry is rugged, such as uncharted seamount, rifts and mid-ocean ridges and continental shelf.*

*[RC3] Line 246. change Polar Regions to polar regions, and elsewhere in the manuscript.*

*[RC3] Line 355 Sentence "The L2 algorithm ....from 15 to 1000 km." is confusing - is this important or not for the ocean?*

*[RC3] line 355 change Hydrology to hydrology*

*[RC3] lines 399-401 Join with preceding paragraph*

*[RC3] Lines 407-411. Join with preceding paragraph.*

*[RC3] Line 410. Change "...dark brown..." to "before calibration (dark brown curve) .."*

*[RC3] lines 412-419 join with preceding paragraph.*

*[RC3] Line 412 change "... orange brown curve ...' to "The residual error (orange brown curve)...."*

*[RC3] Line 412 change "and the light brown curve ....... a factor of 50 for the L3." to "and the residual error after Level-3 calibration (multi-mission, light brown curve), as expected, reduce the ..... a factor of 50 for the L3."*

*[RC3] Line 426 are both figure 10 and figure 11 required to support the noise mitigation section. If not lines 445-455 should be revised in-line with what figure is retained in the manuscript*

*[RC3] Line 464 Change "The green curve is the PSD of the UNet residual..." to "The PSD of the UNet residual (green curve) is more or less ..."*

*[RC3] Move lines 548-561 to follow directly after lines 540. Thus the discussion of features illustrated in figures 13 and 14 are continuous*

*[RC3] Line 550 Figure 13 h is not the same region as figure 14.*

*[RC3] Lines 596-575 Should be joined with previous paragraph*

*[RC3] Combine lines 576-588 into one paragraph*

*[RC3] line 593 change "anyway:" to ", however "*

*[RC3] Line 616 "Change "In panel (a),.... sensors" to "Statistics of SWOT/S3 difference show that all sensors are well behaved (Figure 17 a)."*

*[RC3] Line 637 Change "Then we repeat the process with .. SWOT instruments." to "A similar comparison with Level-3 products of SWOT and Sentinel-3 show that the L3 calibration has significantly improved the consistency with Sentinel-3 for both SWOT instruments (Figure 17 c)."*

*[RC3] lines 695-697 Change to " In this section, we illustrate the utility of our SWOT level -3 product to different research domains. We also discuss limitation of this dataset ...."*

*[RC3] line 700 Change ".... Gulf Stream figures of Figure 1," to ".... Gulf Stream (Figure 1),.."*

*[RC3] Line 701 Change "... mesoscale field is shown in Figure 16" to "...mesoscale field (figure 16).: The entire manuscript should be revised to remove similar writing. Figure are used to support text assertions and referenced within ( ) in most instances.*

*[RC3] Similarly, " Moreover, Figure 13 ..." should be revised in a similar way.*

*[RC3] Figure 4: Add latitude and longitude labels y- and x-axis*

*[RC3] Figure 3. Please label the figure panels. Also put x-, y- longitude and latitude range on outside of plot and add label to the axis.*